



# The ZiCOS-M CO$_2$ sensor network: measurement performance and CO$_2$ variability across Zürich

Stuart K. Grange[1,2], Pascal Rubli[1], Andrea Fischer[1], Dominik Brunner[1], Christoph Hueglin[1], and Lukas Emmenegger[1]

[1]Empa, Swiss Federal Laboratories for Materials Science and Technology, Laboratory for Air Pollution/Environmental Technology, Überlandstrasse 129, 8600 Dübendorf, Switzerland
[2]Wolfson Atmospheric Chemistry Laboratories, University of York, York, YO10 5DD, United Kingdom

**Correspondence:** Stuart K. Grange (stuart.grange@empa.ch); Christoph Hueglin (christoph.hueglin@empa.ch)

**Abstract.**

As a component of the ICOS Cities project, a "mid-cost" NDIR (nondispersive infrared) CO$_2$ sensor network was deployed across Zürich city (Switzerland), known as ZiCOS-M. The network was operational between July 2022 and July 2024 and consisted of 26 monitoring sites, 21 of which were located in or around Zürich city with five sites outside the urban area. Daily
calibrations using two reference gas cylinders and corrections of the sensors' spectroscopic response to water vapour were performed to reach a high level of measurement accuracy. The hourly mean root mean squared error (RMSE) was 0.98 ppm (range of 0.46 and 1.5 ppm) while the mean bias was -0.09 ppm (range of -0.72 and 0.66 ppm) when undergoing parallel measurements with a high-precision reference gas analyser. CO$_2$ concentrations (technically, dry air mole fractions), were highly variable with site means in Zürich ranging from 434 to 460 ppm and Zürich's mean urban CO$_2$ dome was 15.4 ppm
above the regional background. Some of the highest CO$_2$ levels were found at two sites exposed to a combination of strong plant respiration into a very confined nocturnal boundary layer. High CO$_2$ episodes were detected outside Zürich's urban area demonstrating that processes acting on a variety of scales drove CO$_2$ levels. The ZiCOS-M network offered significant insights at an order of magnitude lower cost compared to reference instruments and the observations generated by ZiCOS-M will be used in additional ICOS Cities activities to conduct CO$_2$ emission inventory validation with inversion modelling systems.

## 1 Introduction

### 1.1 Background

Urban areas are very significant sources of atmospheric pollutants and greenhouse gases (GHGs) including carbon dioxide (CO$_2$). In 2020, it was estimated that urban areas were responsible for approximately 70 % of global CO$_2$ emissions (Lwasa et al., 2022). The increased population densities and intensive energy consumption can result in CO$_2$ urban domes, where CO$_2$
is enhanced by a few parts per million (ppm) to tens of ppm in and around the urban extent (Xueref-Remy et al., 2023). The importance of reducing CO$_2$ emissions and the decoupling of carbon emissions from economic growth is a priority for most national and subnational governments, in order to avoid some of the worst negative consequences of anthropogenic climate





change (IPCC, 2023). The importance of $CO_2$ emissions from urban areas has driven top-down analysis methods, where observations of $CO_2$ are combined with atmospheric inversion modelling systems to validate bottom-up emission inventory-based estimates. These two approaches are complimentary and their reconciliation is expected to yield the most reliable emission estimates to allow for potential management.

The monitoring of $CO_2$ in urban areas has generally not been a priority, unlike traditional air pollutants, because of the lack of legal standards. High-quality $CO_2$ time series are generally confined to isolated or remote locations where immediate emission sources are absent. These sites are suitable to capture long-term and large-scale processes, but they are unable to resolve the dynamics of $CO_2$ sources and sinks within urban areas (Hernández-Paniagua et al., 2015). In addition, the technology used for high-accuracy monitoring of $CO_2$ measurement remains expensive (Mao et al., 2012; Martin et al., 2017) and, therefore, the deployment of several $CO_2$ analysers in a city is usually considered cost prohibitive. An alternative approach is to deploy lower cost $CO_2$ sensors and several research groups have deployed monitoring networks in this context (Maag et al., 2018). Although such sensors have lower measurement performance, their poorer accuracy can be offset by being deployed in a larger number and thus, offer the possibility of resolving spatial and temporal patterns at a smaller scale (Peltier et al., 2021). Therefore, the utility of lower-cost sensors can still be high (Bart et al., 2014; Casey and Hannigan, 2018).

Prominent urban $CO_2$ monitoring networks include the Berkeley Environmental Air-quality and $CO_2$ Network (BEACO$_2$N) located across the San Francisco Bay area (Shusterman et al., 2016; Turner et al., 2016; Kim et al., 2018; Delaria et al., 2021), networks in Paris (Arzoumanian et al., 2019; Lian et al., 2024), and the Carbosense network across Switzerland (Müller et al., 2020). The nomenclature regarding the cost points for these networks is inconsistent because the definition of what a lower-cost sensor is varies among operators. Here, we discuss a $CO_2$ sensor network that has been defined as "mid-cost" and is in a price range that is comparable to the BEACO$_2$N and Paris networks. The Carbosense network, in contrast, used sensors at a significantly lower price point, and therefore, would be defined as a low-cost $CO_2$ sensor network.

## 1.2 Switzerland and Zürich

Switzerland is a small country located in Western Europe with a population of approximately 9 million. It is highly developed with a GDP per capita among the highest in the world (International Monetary Fund, 2023). Switzerland has been successful at decreasing its production-based $CO_2$ emissions, especially since 2010 (Ritchie et al., 2020; Federal Office for the Environment (FOEN), 2023). Consumption-based per capita $CO_2$ emissions have not decreased in the same way and remain high reflecting the wealth of the country's residents.

Zürich is Switzerland's largest city and has a population of 430 000 inhabitants in the city proper and another 1 million in the surrounding agglomerations (Stadt Zürich, 2023a). Zürich is located on the Swiss plateau at about 410 m above sea level in an area of complex terrain. Zürich's city centre is situated around Lake Zürich's main northern outflow – the Limmat River – flowing in a northwest direction, which has formed the Limmat Valley where much of the urban area is located. The Limmat Valley is bound to the west by the Albis range and to the east by the discontinuous Pfannenstiel–Altberg hill chain. Zürich's urban area extends beyond the Limmat Valley in all directions, but notably, Districts 11 and 12 are located north of the eastern





hill range and form a continuous urban area over a saddle between Zürichberg and Käferberg. These topographic features of the urban area are relevant for pollutant transport and dispersion processes (Berchet et al., 2017).

Zürich city's local government has legal obligations to be net zero by 2040 regarding direct $CO_2$ emissions and has targets for the reduction of per capita emissions (Stadt Zürich, 2023b). These net zero laws were a result of a Zürich canton referendum in September 2022. The latest emission inventory compiled for 2022 indicates that half of the city's $CO_2$ emissions (51 %) are sourced from stationary combustion, mostly residential and commercial heating emissions with a small contribution from waste incineration (Stadt Zürich, 2024). Public power generation and road transportation are the two other large emission sources (32 and 12 % respectively), and all other sources make up the outstanding 5 %.

### 1.3 ICOS Cities

ICOS Cities[1] is a European Horizon 2020 project that acts as a pilot to test and evaluate different $CO_2$ measurement approaches that provide value for the scientific, policy, and citizen communities within urban areas. Three European cities ranging from large to small – Paris (France), Munich (Germany), and Zürich (Switzerland) are included in the pilot project and all three cities have $CO_2$ sensor networks that form one component of the urban observatories that have been or are deployed in the cities. By design, the three different sensor networks in the three cities differ in their monitoring focus, hardware, and software, but not in their primary objectives. In the Zürich case, the ZiCOS-M network greatly benefited from the experiences gained from the earlier Swiss-wide Carbosense $CO_2$ network (Klose, 2017; Empa, 2019; Müller et al., 2020), but the Carbosense and ICOS Cities activities were separate.

The ZiCOS-M network was designed to supply observations for atmospheric inversion modelling systems, in order to allow for comparisons between top-down estimates of $CO_2$ emissions (natural and anthropogenic) and bottom-up estimates. However, once the sensor network was deployed, it was clear that the network was providing observations at a sufficiently high data quality that were of use for observational analyses of $CO_2$ across Zürich's region and urban area.

### 1.4 Objectives

This work has two overarching objectives. The first is to describe the ZiCOS-M $CO_2$ sensor network's design, deployment, and data processing strategies, and to document what measurement performance was achieved with the sensors. The second is to present the spatial and temporal patterns of $CO_2$ across the network's monitoring domain. The first objective will satisfy two communities: the inversion modelling groups that use the observations generated by the sensor network to evaluate and verify the city's emission inventory, and those who are operating or designing environmental gas sensing networks because many of the data processing approaches are generic and are portable to other networks in other areas and/or networks that target other quantities, such as other greenhouse gases or air pollutants.

The $CO_2$ sensor network deployed across Zürich city for the ICOS Cities project contained two sensor tiers – so-called low- and mid-cost. Here, only the mid-cost sensors' activities and results will be discussed, while the low-cost sensor network

---

[1]https://www.icos-cp.eu/projects/icos-cities





(called ZiCOS-L) will be discussed in a future companion paper because the differing data quality gives rise to challenges when making direct comparisons across the different sensor types, due to their different measurement performances.

At the time of writing, the ZiCOS-M network is still in operation, but imminently, the network will be reconfigured, whereby

half of the monitoring sites will be decommissioned. Therefore, we refer to the ZiCOS-M's operations, sites, and data set in the past tense, except for the more distant background sites that are run by other monitoring activities, because these sites can be considered permanent features.

## 2 Methods

### 2.1 Sensors

The ZiCOS-M network used three different models of NDIR $CO_2$ sensors from three different manufacturers. Most of the sensors (21) were an integration based on the Senseair HPP (high-performance platform) (Senseair, 2016, 2018) sensor (Table 1). The Senseair HPP sensor is a prototype sensor and is no longer in production. These sensors have seen use in similar past ambient monitoring activities in Switzerland (Müller et al., 2020) and related sensors have undergone characterisation elsewhere (Kunz et al., 2018; Arzoumanian et al., 2019; Lian et al., 2024). In addition to the Senseair HPPs, five Vaisala GMP343

(Vaisala, 2013) sensor units were also used, and finally, one Licor LI-850 (LI-COR Biosciences, 2018) was operated in the network. The cost of the sensor units themselves was between CHF/EUR/USD 3000 and 5000, but after integration into a measurement system, the price point was approximately CHF/EUR/USD 5000–10000. The counts of sensors in Table 1 do not include the replacement of the actual $CO_2$ sensors themselves within the sensor packages (which were referred to as sensing elements). Additionally, more sensors were operated than the number of monitoring sites because some sensors failed and were

replaced during the monitoring period.

**Table 1.** The number ($n$) of different sensors or monitors used in the ZiCOS-M sensor network.

| Sensor type | Sensor group | $n$ |
| --- | --- | --- |
| Senseair HPP | $CO_2$ sensor | 21 |
| Vaisala GMP343 | $CO_2$ sensor | 5 |
| Licor LI-850 | $CO_2$ sensor | 1 |
| Picarro G1301 or G2401 | High-precision gas analyser | 4 |
| Decentlab DL-ATM22 | Wind sensor | 14 |

All three sensor models were integrated into a very similar monitoring package by Decentlab GmbH (Decentlab GmbH, 2022), a commercial project partner. Figure 1 shows the main components and configuration of the measurement system. The sensor package included three-way valves for the switching between ambient sampling and two gas cylinders with demand-flow regulators, filters, a sample pump, boards for data acquisition and transmission, temperature and relative humidity sensors installed in the sample gas stream, and for some integrations, ambient air pressure sensors. All hardware, except for the gas




cylinders and demand-flow regulators, was contained in a weatherproof housing (Figure 1; Figure A1). In the case of the Vaisala GMP343 and Licor LI-850 sensors, the ancillary sensors in the gas stream and the ambient pressure sensors were connected to the sensor and onboard compensations for sample temperature, humidity, and ambient pressure were enabled. Originally, it was planned that a consistent data processing approach could be used for the three sensor types, but due to their differing

measurement performance (especially concerning water vapour), different logic was required. Full details of these processes are available in Section 2.3.

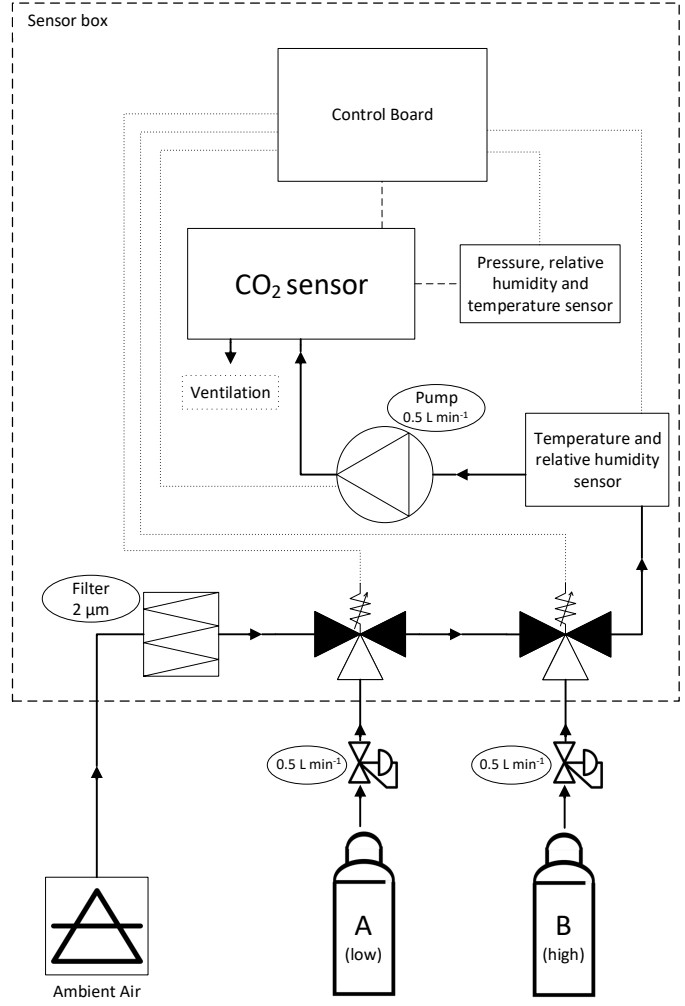

**Figure 1.** Schematic of the sensor measurement system showing the major components of the system and their configuration.

In addition to the $CO_2$ sensors, fourteen sonic wind sensors (Decentlab DL-ATM22) were installed at rooftop sites to provide auxiliary information on air flow and temperatures in the city (Table 1). The sensors' data were transmitted via Swisscom's LoRaWAN network which is a wide area network (WAN) designed for low power applications and small data volumes (LoRa®



Aliance, 2015). The four high-precision gas analysers included in the network's data set were cavity ring-down spectrometers of different generations manufactured by Picarro (Rella, 2010; Rella et al., 2013; Zellweger et al., 2016) that were operated by following routine calibration and data quality control processes.

## 2.2 CO$_2$ monitoring sites

The ZiCOS-M sensor network was composed of a total of 26 monitoring sites with 21 sites in or around the immediate area 125 of Zürich city and five sites in more distant locations. Three of these more distant sites were included in the dataset as they provide critical information on CO$_2$ levels surrounding the city (Figure 2; Table 2) while the other two sites provided CO$_2$ observations from other locations that could be contrasted with CO$_2$ measurements across the Zürich region.

**Table 2.** Basic monitoring site information for the ZiCOS-M sensor network. The elevation represents the elevation above sea level of the monitoring site and the measurement height is the height above ground. The five more distant and background sites outside Zürich are at the bottom of the table.

| Site | Site type | Installation | Measurement height (m) | Latitude | Longitude | Elevation (m) |
|---|---|---|---|---|---|---|
| Albisgüetli | Urban | Rooftop | 22.1 | 47.353 | 8.513 | 470 |
| Badenerstrasse Farbhof | Urban | Rooftop | 22.5 | 47.390 | 8.480 | 400 |
| Bankenviertel Bleicherweg | Urban | Rooftop | 26.5 | 47.369 | 8.538 | 409 |
| Dübendorf-Empa | Urban | Near-ground | 5 | 47.405 | 8.608 | 430 |
| Güterbahnhof | Urban | Rooftop | 29.4 | 47.382 | 8.518 | 408 |
| Hardau II | Urban | Rooftop (high) | 114 | 47.381 | 8.510 | 409 |
| Hardturmstrasse Förrlibuck | Urban | Rooftop | 40.6 | 47.392 | 8.515 | 401 |
| Heubeeribüel | Urban | Near-ground | 1.5 | 47.381 | 8.566 | 615 |
| Kantonales Labor Zürich | Urban | Rooftop | 20.4 | 47.371 | 8.558 | 452 |
| Letzigraben Telefonzentrale | Urban | Rooftop | 24 | 47.379 | 8.501 | 412 |
| Limmattalstrasse Höngg | Urban | Rooftop | 13.5 | 47.404 | 8.488 | 441 |
| Reckenholz | Urban | Near-ground | 4.2 | 47.428 | 8.517 | 443 |
| Schule Milchbuck | Urban | Rooftop | 35.3 | 47.396 | 8.538 | 478 |
| Stauffacherstrasse Werdplatz | Urban | Rooftop | 48 | 47.372 | 8.529 | 411 |
| Tiefenbrunnen Wildbachstrasse | Urban | Rooftop | 38.8 | 47.353 | 8.559 | 409 |
| Universität Zürich Irchel | Urban | Rooftop | 29 | 47.399 | 8.551 | 492 |
| Wollishofen | Urban | Rooftop | 40.6 | 47.347 | 8.533 | 408 |
| Zürich Kaserne | Urban | Near-ground | 3.3 | 47.378 | 8.530 | 409 |
| Rosengartenstrasse | Urban traffic | Near-ground (kerbside) | 2.8 | 47.395 | 8.526 | 433 |
| Schimmelstrasse | Urban traffic | Near-ground (kerbside) | 4.2 | 47.371 | 8.524 | 413 |
| Stampfenbachstrasse | Urban traffic | Near-ground (kerbside) | 4.2 | 47.387 | 8.540 | 440 |
| Beromünster | Rural background | Tower | 212 | 47.190 | 8.176 | 797 |
| Birchwil Turm | Rural background | Tower | 54 | 47.467 | 8.649 | 592 |
| Lägern Hochwacht | Rural background | Elevated | 28 | 47.482 | 8.397 | 845 |
| Sottens | Rural background | Tower | 46 | 46.656 | 6.736 | 775 |
| Jungfraujoch | High-alpine background | High-alpine | 13.9 | 46.548 | 7.985 | 3572 |

The network's testing site, Dübendorf-Empa, is also an air quality morning site that is part of the National Air Pollution Monitoring Network (NABEL) (Empa, 2024). Dübendorf-Empa was used for intercomparison exercises and model training by 130 using the reference CO$_2$ time series provided at this site. Three of the four more distant sites were equipped with high-precision



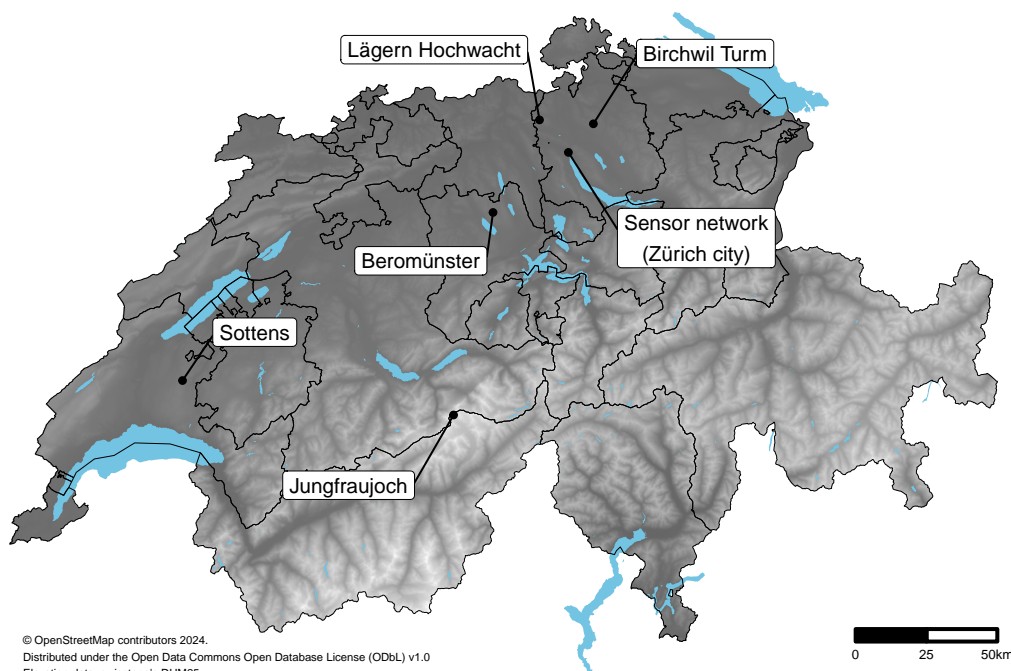

**Figure 2.** Location of the ZiCOS-M sensor network and the more distant locations where $CO_2$ observations were available and used as part of the study's dataset in Switzerland. The internal lines indicate Switzerland's cantonal boundaries and substantial water bodies are also shown.

gas analysers while the fourth site, Sottens (160 kilometres to the south-west of Zürich city), had a sensor that was operated identically to those sensors in Zürich city. The predominant wind directions across the Zürich region are west-south-west and east-north-east, reflecting the orientation of the Swiss plateau. The three background monitoring sites surrounding Zürich city (Figure 2) were positioned in locations that, depending on the wind behaviour, would be down- or up-wind of the city.

The Beromünster background monitoring site is located southwest of Zürich city (Figure 2). It is a tall tower where the sampling system cycles among five different measurement heights. For the dataset presented here, only observations from the highest sampling point at 212 m were used. Lägern Hochwacht is northwest of Zürich city and is located on the forested Lägern hill that is orientated in an east-west direction. Lägern Hochwacht is on the hill's ridge or crest. The sampling height is 32 m above ground which is above the forest's canopy height. For additional details about these two monitoring sites, see

Oney et al. (2015). Birchwil Turm (a telecommunication tower) is another background site located 12.8 km from Zürich's city centre in a north-east direction next to an electrical substation in the Zürcher Unterland (Table 2; Figure 3). Birchwil Turm is approximately halfway between Zürich city and Winterthur, Canton Zürich's second largest city. Jungfraujoch in the Bernese Alps, 102 km from Zürich at an altitude of 3572 m, was used as the study's European or hemispheric background site (Figure 2). Jungfraujoch is an observatory that includes NABEL and ICOS activities. The $CO_2$ time series from this location serves as a



reference to represent background European $CO_2$ with an absence of any immediate significant emission sources (Pieber et al., 2022).

The monitoring sites within and around Zürich city were classified further by their installation or siting types. The majority of sensors (14 of the 22 sensor sites) were deployed with inlets sampling at rooftop level (Table 2; Figure 3; Figure A1). The sensors were generally installed in maintenance rooms reserved for mobile phone infrastructure, and for most installations, 150 these rooms were temperature-controlled. The focus on rooftop installations was done to represent $CO_2$ emission sources in the city without being primarily forced by immediate emission sources as would occur with sampling points at ground level in street canyon environments. The rooftop sites offer spatial representativeness across the city at a resolution that was optimised to the mesoscale model systems' spatial resolution. The sites were carefully selected out of a large number of potential locations by requiring minimal impact from nearby ventilation or heating stacks that were present on many roofs.

Hardau II is a central monitoring site of the network, which not only features a mid-cost sensor but also an Eddy covariance system to directly measure the $CO_2$ fluxes in a central part of the city. Measurements are conducted at 114 m above ground on a 95.3 m tall building, which is much higher than other rooftop sites. Several extra monitoring activities were conducted at this site during an intensive campaign between September 2022 and March 2023 within the ICOS Cities project. Details on the other monitoring activities at Hardau II are available in Stagakis et al. (2023).

**2.3 Data**

The principal steps of the extensive data processing for the ZiCOS-M network are outlined and explained below. The data processing logic was dependent on the sensor type (Section 2.1) because of their variable performances relating to what onboard correction algorithms were activated. A schematic of data processing steps is shown in Figure A2 and the main equations are presented in the appendix. The final set of observations was classified as Level-2A following the processing levels proposed 165 by Schneider et al. (2019).

At a high level, the data processing steps can be grouped into three operations: (*i*) the collection and formatting of data from different sources to conform to a formal data model and framework for convenient access and interaction, (*ii*) the application of various adjustment strategies to improve the measurement quality (*i.e.*, to increase their agreement with observations generated by reference instrumentation) without moving to outright model predictions (Schneider et al., 2019), and (*iii*) the handling and 170 integration of metadata units such as when and where sensors were located, calibration gas information, and what observations have been invalidated and why. All data processing was conducted with the R programming language (R Core Team, 2023). The database technology used was PostgreSQL (PostgreSQL Global Development Group), and the formal time series relational data model was an extended **smonitor** data model previously developed for air quality applications that overlaps with greenhouse gas monitoring almost completely (Grange et al., 2017; Grange, 2018, 2019a, b).

**2.3.1 Transmission and storage**

All sensor data were transmitted through the LoRaWAN network to Decentlab's cloud storage infrastructure (Decentlab GmbH, 2022) where the data packets were decoded and made accessible via a simple application programming interface (API) (Grange,



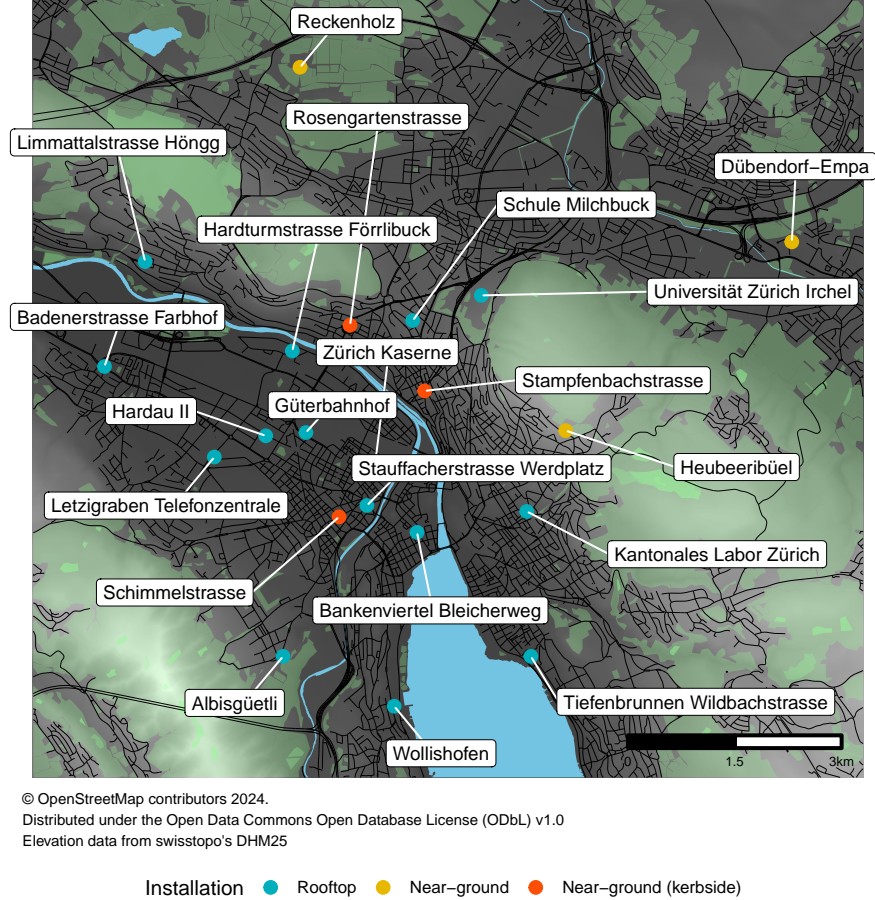

**Figure 3.** The ZiCOS-M $CO_2$ sensor sites (additional details can be found in Table 2) in and around the vicinity of Zürich city. Vegetated areas, terrain, and substantial water bodies are also shown.

2024b). Depending on the sensor type, diagnostic variables along with raw measurement values (the IR signal in the specific case of NDIR sensors), ancillary data such as temperature, relative humidity, and pressure from other sensors integrated into the sensor product as well as onboard calculated $CO_2$ concentration were transmitted and stored. In the **smonitor** nomenclature, the $CO_2$ calculated by onboard algorithms (that are generally proprietary) and coefficients determined by factory calibration was called *reported $CO_2$*. The sensors reported observations approximately every 60 seconds; however, because of limitations of the LoRaWAN protocol, usually 57–58 measurements were successfully transferred and stored per hour.

Several other data sources were accessed in an automated or semi-automated fashion. These data sources include: observations from a $CO_2$ analyser installed at the Dübendorf-Empa NABEL monitoring site (Figure 3), which was the facility used for field testing and parallel measurements, observations for two additional high-precision $CO_2$ analysers in Switzerland but outside Zürich city (Lägern Hochwacht and Beromünster; Figure 2), observations for three NABEL monitoring sites that hosted meteorological instrumentation, two MeteoSwiss data sources – the VQEA33 data product that contains MeteoSwiss's 'core'



sites or stations that host a full suite of meteorological sensors (five sites in and around Zürich were available), and observations

sourced from the IDAWEB portal (MeteoSwiss, 2009) for five additional meteorological sites in and around Zürich city, and finally Jungfraujoch's high-alpine (at 3572 m altitude; Figure 2) validated and near-real time $CO_2$ observations from the ICOS Carbon Portal (Emmenegger et al., 2023, 2024). All of these data sources were integrated into the common data model and stored for uniform access and interaction.

### 2.3.2   Water dilution effect correction and dry air mole fractions

Atmospheric $CO_2$ is usually reported as dry air mole fraction (in units of $\mu mol\,mol^{-1}$ or parts per million, ppm), because this quantity is preserved, not only during atmospheric transport, but also under processes changing the moisture content of the air (Tans and Thoning, 2020). Here, we often use the more generic term "concentration" interchangeably, but in all cases beyond reported $CO_2$, the more accurate definition of dry air mole fraction is correct.

All NDIR $CO_2$ sensors used in the network reported $CO_2$ in moist air. To convert to dry air mole fractions, an estimate of the

water vapour in the gas sample was required, which was obtained from the ancillary temperature and relative humidity sensors installed in the gas stream and the ambient pressure sensors installed within the sensors' waterproof box. The vapour pressure of water was calculated from the relative humidity and the saturation vapour pressure according to the August-Roche-Magnus equation (Alduchov and Eskridge, 1997; Lawrence, 2005). This conversion to dry air mole fractions is referred to as water dilution correction in the following text.

The Senseair HPP sensors also required their reported $CO_2$ to be explicitly normalised to standard atmospheric pressure (1013.25 hPa) before the dilution correction. This was necessary because a clear $CO_2$ dependence on ambient pressure was observed during testing, which indicated that the integrated pressure sensor was not properly used for onboard pressure normalisation. This extra transformation was not required for the two other sensor types as they did not show a dependence on air pressure during the same tests.

### 2.3.3   Reference gas cylinder calibrations

The sensors were integrated with three inlets, an inlet for ambient air sampling and two inlets for the connection of two reference gas cylinders containing known $CO_2$ traceable to the WMO $CO_2$ X2019 calibration scale (Hall et al., 2021). Across the network, the sensors were deployed with 5 or 10 L 'high' and 'low' reference gas cylinders ($\approx 400$ and $\approx 600$ ppm, respectively; Figure 1). The inlets were switched from ambient to low and high inlets (in that order) every 25 hours for 10 minutes. The

first preceding and four subsequent minutes before and after the gas tests were handled as a non-ambient sample to ensure the sample system was flushed of cylinder-sourced gases and did not contaminate the ambient samples.

For each gas test, the period was isolated, the first and final three minutes were discarded and the median was taken to represent the sensor's $CO_2$ for the cylinder test. This trimming and summary logic robustly captured the $CO_2$ concentration in the gas stream from the cylinder after the sensor had reached stability and the gas stream's flow, pressure, and humidity had the

opportunity to equilibrate during the test period. Because the calibration gases were nearly dry, a gas sample humidity criterion





of less than 10 % relative humidity was applied to determine whether the test was valid. A moist sample indicated a leak or an empty cylinder. All test summaries were stored for later use.

The high and low test summaries were used to compute a slope and offset with simple linear regression – albeit (usually) with only two points. The final calibrated dry air mole fraction was then computed from the reported dilution-corrected dry air
mole fraction (Figure A2).

Infrequently, a gas test summary was correctly calculated, but the result was a clear outlier. This was usually driven by poor data capture or bad valve articulation during the gas test. Therefore, the gas test summaries were passed through an interquartile range filter to remove these outliers. Three-day rolling means of the slope and offset coefficients were calculated to slightly smooth the coefficients and to avoid a relatively large change in a coefficient occurring when traversing the midnight boundary
during the moving from one calendar day to the next (Figure A3). If a daily test was missing due to operational issues, the application of a last observation carried forward process was applied to the coefficients.

All sensors were able to be successfully corrected for their relatively constant change in baseline and sensitivity over the monitoring period with the daily reference gas tests (an example is shown in Figure A3). The remaining variability may be partly explained by spectroscopic effects driven by various changes in environmental conditions, such as temperature and
pressure. When there is access to two-cylinder gas tests for such an adjustment, other strategies are also possible; for example, to use the low gas for an offset adjustment and the high gas as a target gas for quality control purposes. However, since the sensors did not only reveal a drift in offset but also gradual changes in sensitivity, it was necessary to calibrate both offset and slope to meet the performance objectives of an uncertainty of about 1 ppm.

### 2.3.4 Correction of the water-induced response

During intercomparison exercises, a measurement performance issue was uncovered with some of the sensors, specifically the Senseair HPP sensors, the sensor model that was most frequently used in the network (Table 1). The issue was identified as a response to water vapour that was not related to the dilution effect. For NDIR measurement technologies and especially for the measurement of $CO_2$, this feature is known and can be potentially corrected for by the sensor-internal data processing (McDermitt et al., 1993; LI-COR Biosciences, 2023). We labelled the effect as a generic *water-induced response* because
the exact mechanism was not confirmed, but will most likely include a combination of spectroscopic features including band broadening, crosstalk, and/or interference. This issue manifested not only in suboptimal performance in ambient monitoring but also as a positive bias of a few ppm, despite the reference gas calibrations. A bias of this magnitude was problematic and was caused by the reference gases in use being dry. The reference gases were thus completely absent of the interferent that was present during ambient monitoring.

The water-induced response was quantified for each sensor using the following laboratory setup. A spiral sampling line with a length of one metre complete with fittings to allow for connections with cylinders containing known $CO_2$ and water injections. The gas cylinder was connected to the sampling line and after a settling period, 0.5 mL of Milli-Q water was injected into the sampling line and sealed. The gas was sampled at a flow rate of 0.5 L min$^{-1}$. The experiment lasted for about 90 minutes, during which the humidity decreased from approximate saturation to nearly zero. This test was repeated with three reference gases



containing 418, 492, and 614 ppm $CO_2$. The obtained data were used to train a multiple linear regression model with two terms, absolute humidity and $CO_2$, which was later used to correct for the water-induced response. The humidity tests showed that the factory correction for the water-induced response was adequate in the case of the Vaisala GMP343 sensor type (Figure A4); hence, this correction was not implemented for the five sensors that were used in the network. The water-induced response is not necessarily a linear function of humidity and $CO_2$ (Wu et al., 2023). However, our tests showed that a linear fit was

sufficient and consistent with the overall performance of the sensors (presented and discussed in Section 3.1).

### 2.3.5 Flagging observations for possible local contamination

A challenging issue was the handling of possible contamination from sources in the immediate vicinity of the sensors' sample inlets. This is a general difficulty for monitoring in urban areas with their numerous sources, and it was especially clear for some rooftop monitoring locations where the inlets were located in close proximity to stacks for heating and/or hot water

gas-fired boilers and ventilation shafts. Although an observation taken in such a situation is valid in an operational sense, for some analyses, such measurements need to be removed or handled in another way. Therefore, an algorithm was implemented to detect potential local contamination. The Hampel identifier was used, which is a windowed median absolute deviation (MAD) filter where a value's distance from the median is evaluated (Pearson et al., 2016), and if that value is outside a threshold, then it is identified as an outlier. The Hampel identifier works in the same manner as other windowed filters, such as those documented

and defined as 'despiking' algorithms by El Yazidi et al. (2018), and it was very effective at identifying local contamination events in the rooftop-sampled $CO_2$ time series as in, for example, Figure A5.

The flagging was applied to all 14 sites that were classed as rooftop sites (Table 2). The median potential local contamination fraction among these sites was 0.9 % but one site, Limmattalstrasse Höngg, had a much greater fraction of 5.1 % (Table A1). For the analyses presented here, all observations that were classed as potentially contaminated were removed. All time series

were also manually inspected as part of the network operations. Erroneous measurement spikes and observations taken during maintenance activities were also flagged to ensure that downstream data users had information on the validity of any given observation.

## 3 Results and discussion

### 3.1 Sensor $CO_2$ measurement performance

Measurement performance of all sensors was assessed during ambient measurements carried out at the Dübendorf-Empa air quality monitoring site (Table 2; Figure 3), which was equipped with a calibrated high-precision $CO_2$ analyser (a Picarro G1301) and produced the reference $CO_2$ time series that was considered a ground truth. The objective of the intercomparisons was to independently test the sensor system in conditions that were very similar to what the sensors experienced in field deployment. In our case, this was with an inlet sampling ambient air and the sensor installed inside, usually in an air-conditioned




room. The analyser and sensors were run independently of one another, and their observations were only compared after the monitoring period. Therefore, the intercomparisons were classed as independent, parallel measurements.

The targeted minimum duration of the parallel measurements was two weeks (336 hours), but generally, this period was longer. The data processing was undertaken in the way it would be conducted in the field, with the same logic being applied for dry air mole fraction transformation, the subtraction of water-induced response if required, and the calibration of observations

based on the daily cylinder tests (Section 2.3). The facility did not have the physical space to test all sensors at the same time, so the sensors were tested sequentially or in pairs over a period of 12 months. Therefore, the sensors experienced different environmental conditions from one another during testing, which could explain some of the inter-sensor variation observed. Standard pairwise performance statistics were computed. The equations and descriptions of these statistics can be found in Peters et al. (2022).

The sensors' average hourly mean root mean squared error (RMSE) was 0.98 ppm and ranged between 0.46 and 1.5 ppm, depending on which sensor was under test (Figure 4; Table A2). The average mean bias was -0.09 ppm, but the average mean bias also demonstrated values between -0.72 and 0.66 ppm due to sensor variation. The calculation of the RMSE error statistic, which covers both systematic (bias) and random components, can be interpreted as the average uncertainty of the estimators' (the sensors') predictions. The measurement uncertainty of approximately 1 ppm, as determined during testing, can

be confidently translated to field conditions because of the design of the testing procedure.

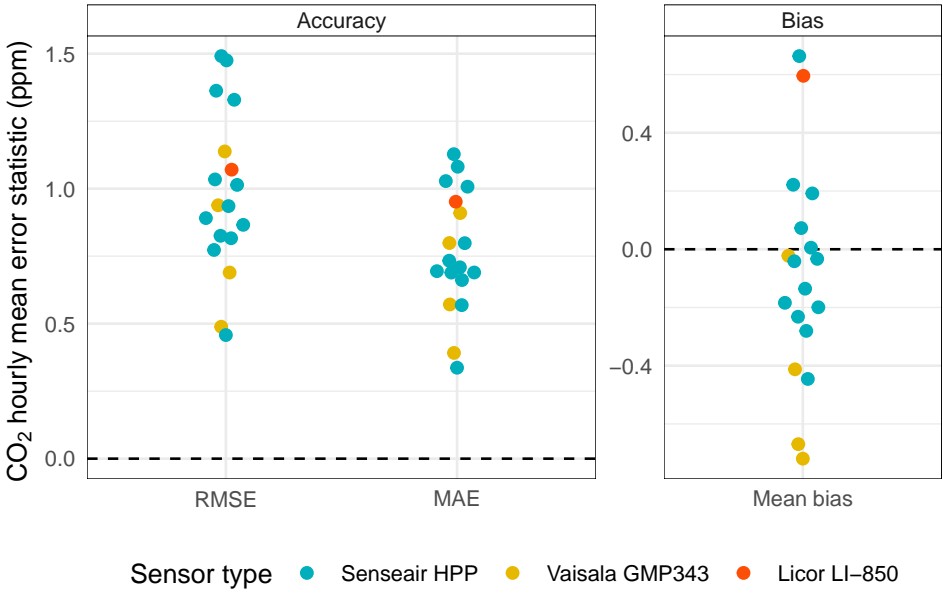

**Figure 4.** Hourly $CO_2$ error statistics of the sensors that were exposed to parallel measurements with a high-precision reference gas analyser at the Dübendorf-Empa monitoring site.

The discontinued Senseair HPP sensor often performed poorer than the commercially available Vaisala GMP343 and Licor LI-850 sensors. The inter-sensor variability of the Senseair HPP sensors was also higher, with some sensors displaying more



measurement uncertainty than others (Table A2). This variation may have been partially caused by the variability of age or uptime because some of these sensors had been used previously for monitoring in other studies (see Müller et al., 2020) or
305 differing environmental conditions during the testing period. Furthermore, the Senseair HPP sensor was a prototype, possibly resulting in larger variations in the product. However, because of the irregular number of each sensor type, additional sensor units would need to be used to confirm these apparent patterns.

Scatterplots of hourly means (Figure 5) demonstrated that all three sensor types displayed excellent $CO_2$ measurement quality and linearity across the detected ambient $CO_2$ range. The scatterplots also show the importance of data processing
and the improvements in measurement performance in both bias and dispersion dimensions, that were achieved due to the application of correction and calibration processes (moving sequentially from the top to bottom panels in Figure 5). In general, the Senseair HPP sensors displayed more dispersion around the reference observations, and therefore, a larger measurement error than the other two sensor models. A somewhat surprising observation was that, despite the Licor LI-850 measuring and reporting water vapour directly, a small water-induced response was still experimentally confirmed. Achieving the measurement
performance shown in Figure 4, Table A2, and Figure 5 was only possible with these NDIR sensors with the utilisation and careful handling of reference gases, which is not feasible for many other atmospheric gases – including the family of reactive gases (and particulates) that are important for air quality.

The use of reference gases increased the complexity of the measurement system and the maintenance required for the operation of the network. However, the gases were critical for achieving the reported measurement performance (Figure 5).
An alternative strategy would be to use a single gas and apply only an offset correction. A sensitivity analysis exploring this approach was conducted by withholding the 'high' gas tests and only using the 'low' $CO_2$ cylinder ($\approx 400$ ppm) for an offset correction. The mean RMSE and bias penalties of this approach were 1.7 and 0.8 ppm and the corresponding medians were 0.17 and 0.12 ppm (compared to the two-gas calibration discussed above). The median penalties of using a single reference gas were rather low and may be acceptable for some monitoring applications, but the larger means demonstrate that for some of
the sensors (three in this case) the sensitivities altered during the testing procedure, and thus, the second gas test was required to address this robustly. This sensitivity analysis showed that an offset correction strategy could be used in some scenarios, but the sensors' sensitivity stability should be regularly tested to ensure that large measurement performance penalties do not arise.

The intercomparison periods were also used to evaluate the effectiveness of the water vapour correction that was determined in laboratory tests under real conditions. Figure 6 shows the response of the three $CO_2$ sensor types to absolute humidity during
their intercomparison periods. Without correcting for the water dilution effect, the values were biased low in comparison to the reference because the sensors reported $CO_2$ in moist air. In the case of the Senseair HPP, converting from moist to dry mole fractions (*i.e.* correcting for the water dilution), led to an overestimation of $CO_2$. Additionally, correcting for the water-induced response brought the residuals close to zero, demonstrating the importance of this additional correction.

An hourly $CO_2$ measurement uncertainty of 1 ppm is comparable to other studies that report such results for $CO_2$ NDIR
sensors (Shusterman et al., 2016; Martin et al., 2017; Kunz et al., 2018; Arzoumanian et al., 2019; Müller et al., 2020; Delaria et al., 2021; Lian et al., 2024). Direct comparisons among the various studies are difficult due to the different endpoints that were defined, variable testing or intercomparison designs, and if the stated uncertainty or error truly represents field conditions.





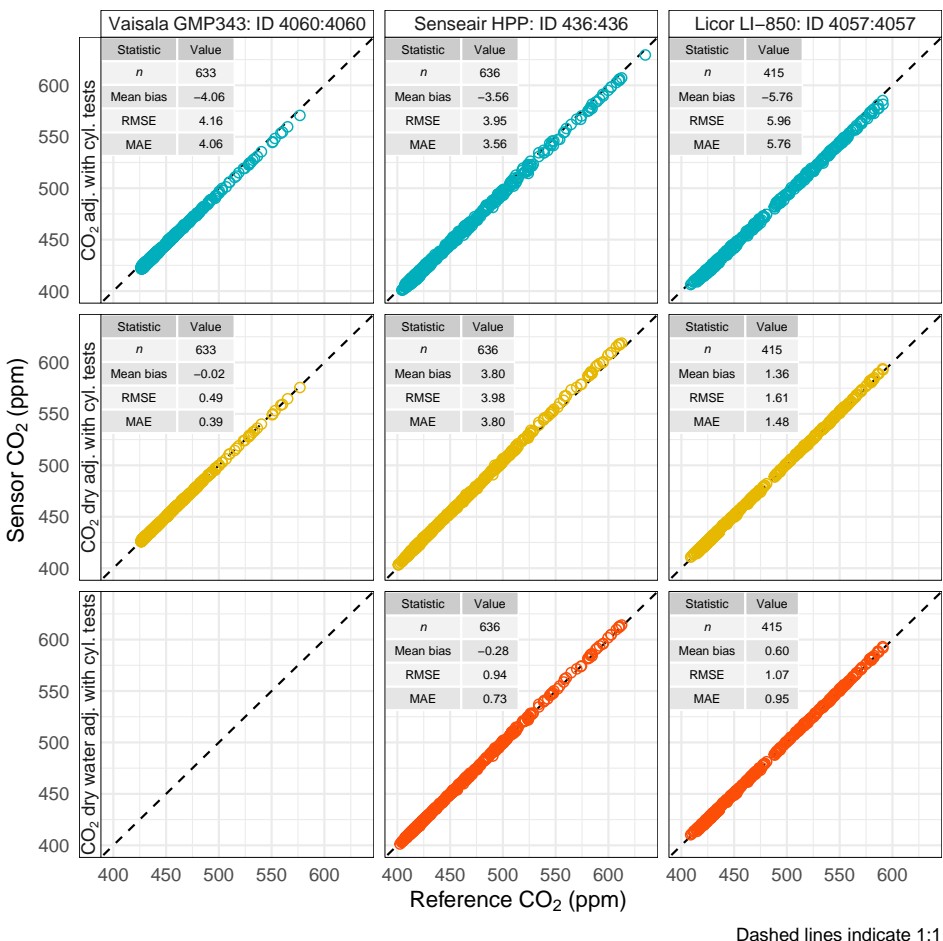

Dashed lines indicate 1:1

**Figure 5.** Hourly $CO_2$ means for selected sensors for the three sensor models used in the ZiCOS-M network during intercomparison at the Dübendorf-Empa monitoring site. The improvements in measurement performance can be seen when moving from top to bottom as the data processing adjustments are applied. The Vaisala GMP343 sensors did not require a water-induced adjustment, therefore, this was not calculated and is not shown.




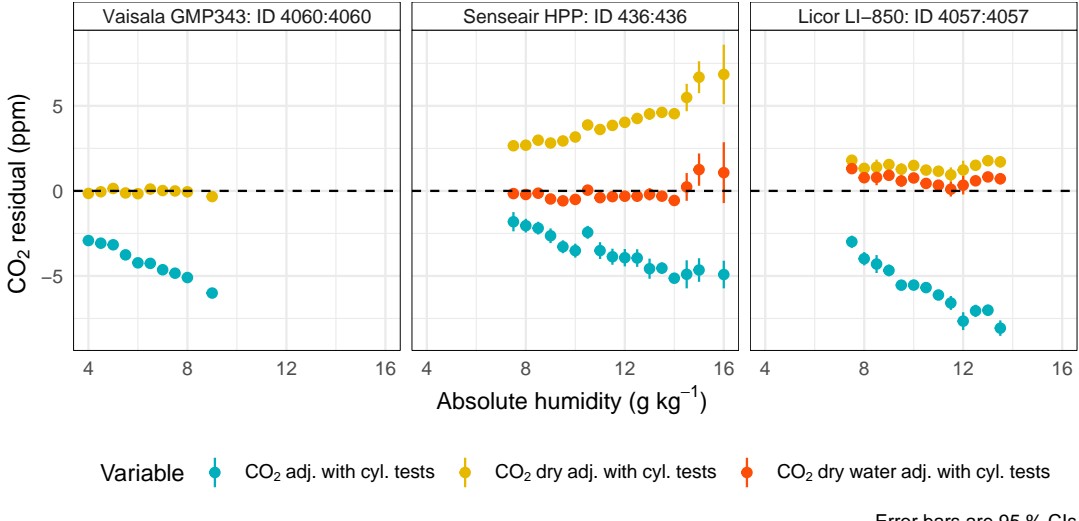

**Figure 6.** $CO_2$ residuals by sample absolute humidity (in $0.5\,\mathrm{g\,kg^{-1}}$ bins) during parallel measurements with a high-precision reference gas analyser at the Dübendorf-Empa monitoring site. The clear influence of dilution (negative) and water-induced response (positive) can be observed in the Senseair HPP example, as can the efficacy of the post-measurement adjustments or corrections.

However, because many studies have reported rather similar results, it is likely that measurement uncertainties of $\approx 1$ ppm can be expected with the current generation of $CO_2$ NDIR sensor products. In contrast, the World Meteorological Organization

has set compatibility goals of 0.1 and 0.05 ppm for background monitoring sites in the northern and southern hemispheres, respectively, when using state-of-the-art monitors (World Meteorological Organization, 2014). Clearly, these types of sensors are unsuitable for background monitoring sites at this time.

## 3.2  $CO_2$ variation in and around Zürich city

### 3.2.1  Site means

The ZiCOS-M sensor network that included 26 monitoring sites demonstrated that $CO_2$ was highly variable across time and the Zürich area during the monitoring period between July 2022 and July 2024. The sensor network's urban background site for the Zürich region, Birchwil Turm (Figure 3), experienced a mean $CO_2$ of 434 ppm while the highest mean $CO_2$ concentration of 460 ppm was found at Rosengartenstrasse (Table 3; Figure 7). Reckenholz observed the second highest mean $CO_2$ and notably was a rural, near-ground monitoring site with an inlet height of 4.2 m (Table 3). The high levels of $CO_2$ were not driven

by anthropogenic emission activities, but by strong biogenic respiration in the early hours of the morning. Dübendorf-Empa, the network's testing site was also exposed to strong biogenic forcing with peak $CO_2$ occurring in the early morning during the growing seasons including spring, summer, and autumn. The processes governing these features are further explored in Section 3.2.2.





**Table 3.** Summary statistics for the $CO_2$ monitoring sites between between July 2022 and July 2024. $CO_2$ unit ppm representing dry air mole fractions and the sites are ordered by their mean $CO_2$.

| Site | Site type | Installation | Mean | Median | Min. | Max. | Data capture (%) |
|---|---|---|---|---|---|---|---|
| Jungfraujoch | High-alpine background | High-alpine | 421.4 | 421.9 | 407.0 | 460.5 | 94.0 |
| Beromünster | Rural background | Tower | 423.4 | 423.5 | 401.8 | 466.1 | 60.9 |
| Lägern Hochwacht | Rural background | Elevated | 427.7 | 427.7 | 401.0 | 473.4 | 93.3 |
| Sottens | Rural background | Tower | 427.8 | 427.5 | 404.6 | 477.4 | 96.5 |
| Birchwil Turm | Rural background | Tower | 433.6 | 432.4 | 401.4 | 496.8 | 91.9 |
| Albisgüetli | Urban | Rooftop | 438.4 | 433.9 | 399.0 | 562.3 | 96.8 |
| Heubeeribüel | Urban | Near-ground | 438.7 | 436.5 | 400.6 | 577.9 | 94.9 |
| Hardau II | Urban | Rooftop (high) | 441.0 | 435.3 | 404.1 | 603.7 | 98.1 |
| Universität Zürich Irchel | Urban | Rooftop | 442.2 | 436.3 | 402.8 | 627.8 | 95.3 |
| Tiefenbrunnen Wildbachstrasse | Urban | Rooftop | 444.2 | 438.4 | 400.6 | 649.8 | 96.3 |
| Kantonales Labor Zürich | Urban | Rooftop | 444.4 | 440.2 | 402.5 | 649.1 | 94.3 |
| Wollishofen | Urban | Rooftop | 444.8 | 439.3 | 403.2 | 611.4 | 95.8 |
| Letzigraben Telefonzentrale | Urban | Rooftop | 446.7 | 439.4 | 401.7 | 603.9 | 98.4 |
| Schule Milchbuck | Urban | Rooftop | 447.4 | 440.7 | 402.6 | 616.1 | 86.0 |
| Stauffacherstrasse Werdplatz | Urban | Rooftop | 447.5 | 440.4 | 402.3 | 646.8 | 97.5 |
| Hardturmstrasse Förrlibuck | Urban | Rooftop | 449.0 | 440.6 | 402.6 | 599.6 | 87.8 |
| Güterbahnhof | Urban | Rooftop | 449.7 | 441.8 | 405.9 | 624.5 | 96.6 |
| Zürich Kaserne | Urban | Near-ground | 449.8 | 443.0 | 403.3 | 636.3 | 97.1 |
| Bankenviertel Bleicherweg | Urban | Rooftop | 449.9 | 443.4 | 407.9 | 638.4 | 88.9 |
| Limmattalstrasse Höngg | Urban | Rooftop | 450.7 | 443.8 | 403.3 | 599.3 | 84.5 |
| Badenerstrasse Farbhof | Urban | Rooftop | 451.3 | 444.1 | 402.2 | 615.3 | 79.3 |
| Stampfenbachstrasse | Urban traffic | Near-ground (kerbside) | 451.6 | 444.4 | 405.2 | 647.9 | 94.1 |
| Dübendorf-Empa | Urban | Near-ground | 457.6 | 443.2 | 400.9 | 646.3 | 98.3 |
| Schimmelstrasse | Urban traffic | Near-ground (kerbside) | 458.6 | 450.8 | 407.7 | 646.3 | 97.4 |
| Reckenholz | Urban | Near-ground | 459.4 | 441.4 | 399.0 | 732.4 | 94.4 |
| Rosengartenstrasse | Urban traffic | Near-ground (kerbside) | 460.3 | 453.4 | 409.7 | 643.7 | 92.5 |

Rosengartenstrasse was the most enhanced monitoring site with respect to $CO_2$. Rosengartenstrasse is in Zürich city proper
and is located in a kerbside environment next to an uphill and northbound section of an arterial road. It displayed a clear,
traffic-forced diurnal cycle of $CO_2$ during the day, as observed in other cities (Shusterman et al., 2018). The two other traffic
sites located in kerbside environments – Schimmelstrasse and Stampfenbachstrasse – also experienced some of the highest
$CO_2$ of the network (Table 3; Figure 7).

Site means for the daytime hours (local time between 10:00 and 16:00) are also provided in Figure A6. The gradient between
background and rooftop sites was, on average, about 25 ppm for daily means but only 15 ppm for daytime means. For the
kerbside locations Rosengartenstrasse and Schimmelstrasse the gradient was about 30 ppm in both cases suggesting that their
concentrations are primarily determined by their proximity to traffic, and less so by atmospheric dispersion dynamics.




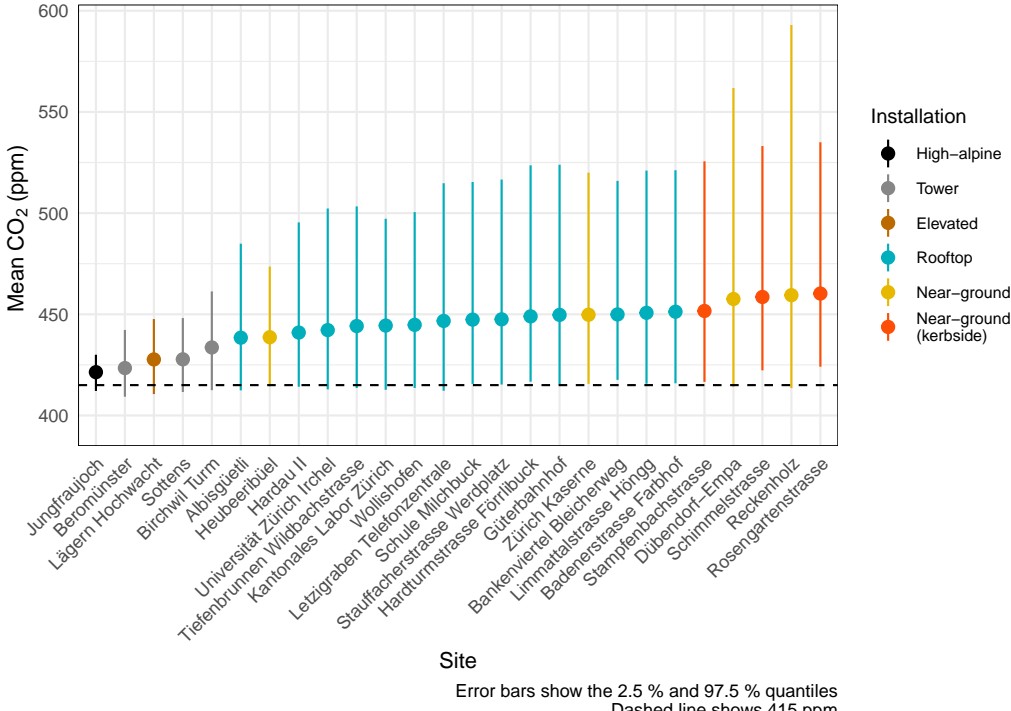

**Figure 7.** Mean $CO_2$ for ZiCOS-M's 26 monitoring sites between July 2022 and July 2024.

When considering only the rooftop sites with measurement heights between 13.5 and 114 m (Table 2), the daily means ranged from 438 ppm at Albisgüetli to 451 ppm at Badenerstrasse Farbhof (Table 3), a mean gradient within the city of 13 ppm.

For daytime means the gradients were only about half this magnitude, but they were still sufficiently large to be reliably captured by the sensors, given their measurement uncertainty of 1 ppm (Section 3.1). Jungfraujoch, the high-alpine monitoring site located in the distant Bernese Alps at an elevation of 3572 m, experienced a daily mean of 421 ppm which represented European background $CO_2$ during the period of monitoring. Another interesting observation is that the distant sensor site in Sottens in Vaud (western Switzerland; Figure 2) had lower mean and median $CO_2$ than Zürich's regional background site

Birchwil Turm, despite their comparable installation and sampling height, which suggests the source-sink dynamics across the Swiss plateau were variable during the monitoring period.

### 3.2.2 Diurnal cycles and ranges

The diurnal cycle of $CO_2$ at the 26 sites in the ZiCOS-M network formed three broad groups reflecting different source and sink dynamics that can be classified as background, anthropogenic-influenced, and biogenically forced. Background sites

demonstrated minor changes in mean hourly $CO_2$ throughout the year but, except for Jungfraujoch, small amounts of morning $CO_2$ enhancement and afternoon reduction were present in all seasons (Figure 8). The magnitudes of the morning $CO_2$ peaks



were higher than those experienced in the afternoon or evening. Three-monthly definitions of seasons were used, where summer refers to the months of June, July, and August.

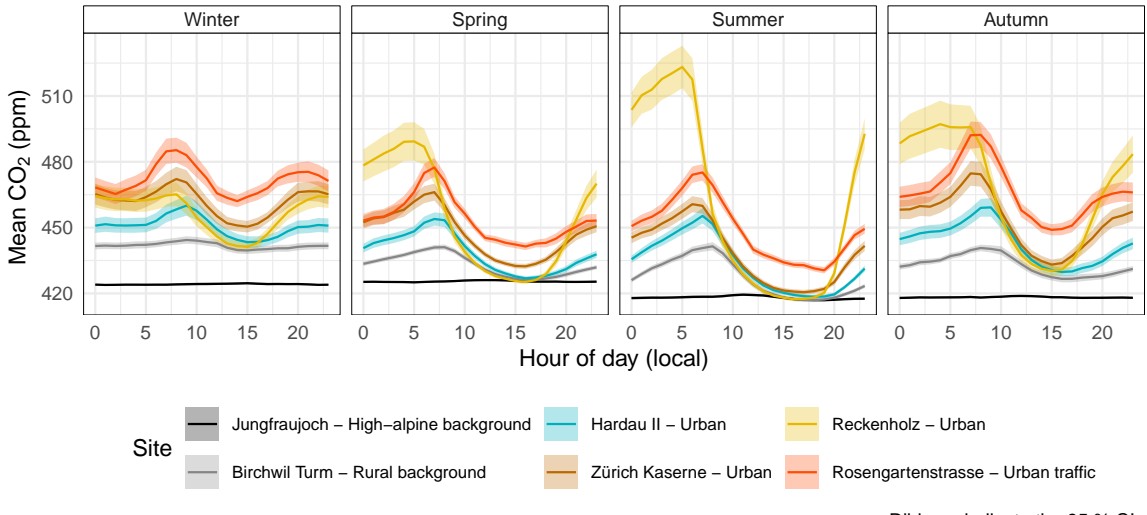

**Figure 8.** Seasonal $CO_2$ diurnal cycles for selected monitoring sites with different site type classifications between July 2022 and July 2024.

Most sensor sites were located in or around Zürich's urban area, and they showed an anthropogenic-influenced diurnal cycle.
These anthropogenic-influenced sites were distinguished by $CO_2$ levels peaking in the morning (usually between 06:00 and 10:00), driven by traffic and other anthropogenic emission processes at these times. A combination of increased atmospheric dispersion and biogenic uptake in the early and mid-afternoon reduced $CO_2$ to the daily minima (Figure 8). During summer afternoons, many of the anthropogenic-influenced sites' mean $CO_2$ were below the European background at Jungfraujoch, due to uptake by the local and regional biosphere, reflecting the ground-level acting as a net sink in this season (Stephens et al.,
2007). Outside the growing season, *i.e.*, winter, the anthropogenic-influenced sites showed a clear second peak in the afternoon or evening due to traffic and heating emissions. This was reminiscent of the diurnal cycles observed for primary air pollutants because they are co-emitted from the same sources (West et al., 2013; Fiore et al., 2015; The Royal Society, 2021), and due to the lack of strong active $CO_2$ sinks at this time of the year.

The Reckenholz and Dübendorf-Empa sites formed the third, biogenically forced, diurnal cycle group. This group was
strikingly distinct from the other groups and was identified by peak $CO_2$ being reached in the early hours of the morning (04:00–06:00) during the growing seasons (Figure 8). These early morning $CO_2$ peaks could exceed 730 ppm which was higher than peak $CO_2$ observed at the three kerbside monitoring sites. A combination of $CO_2$ emissions from ecosystem respiration processes into an extremely confined nocturnal boundary layer with no or very little advection caused these rather extreme $CO_2$ levels. These observations were clear demonstrations of what is known as the rectifier effect where boundary
layer dynamics and $CO_2$ fluxes are temporally correlated, which amplifies the diurnal variability of $CO_2$ beyond the magnitude expected from source-sink dynamics alone (Denning et al., 1999; Shi et al., 2020). Similar observations, such as those captured





by ZiCOS-M, have been made previously, albeit somewhat less pronounced, in Switzerland (Gimmiz) (Oney et al., 2015) and Canada (Vancouver) (Crawford and Christen, 2014). Both Reckenholz and Dübendorf-Empa have significant amounts of forest and crop fields in the surrounding area and furthermore, Reckenholz is located in a shallow depression. For all three diurnal

cycle groups, the daily $CO_2$ minima were always reached in the afternoon (between 14:00–16:00 local time) and showed little temporal variation among the different seasons.

To further characterise the diurnal variability across the network, the amplitude of the mean differences between daily minimum and maximum hourly means were calculated for each season and all sites. The two biogenically forced Reckenholz and Dübendorf-Empa sites showed large diurnal ranges in all seasons, apart from winter, with diurnal ranges peaking at 106

and 81 ppm in the summer months (Figure 9). Diurnal ranges of this magnitude were greater than those reported in previous studies, for example, Vogt et al. (2006). The extremely high values in the early morning hours are responsible for these two sites' high mean $CO_2$ values, presented in Figure 7 and Table 3. The high-alpine Jungfraujoch site is largely isolated from localised $CO_2$ sources and sinks, and thus showed almost no diurnal range throughout all four seasons. This contrasted with all other sites in the network, which experienced their most pronounced diurnal cycle in summer, followed by spring or autumn

due to the correlated diurnal variation in boundary layer depth and natural and anthropogenic sources and sinks during these seasons. The low diurnal ranges observed in winter can be mostly explained by only anthropogenic sources being active, because the climate of Zürich is such (Köppen-Geiger climate classification: *Cfb* (Beck et al., 2018)) that plant respiration and photosynthesis drops to near-zero for much of this period (Zubler et al., 2014).

### 3.2.3 Quantification of Zürich's urban $CO_2$ dome

The ZiCOS-M network was used to calculate overall and seasonal environmental increments, notably the regional $CO_2$ enhancement of the Swiss Plateau compared to European background mole fractions represented by Jungfraujoch and the magnitude of Zürich city's urban $CO_2$ dome, with respect to the regional background. Between July 2022 and July 2024, Zürich's regional background was on average 9.3 ppm larger than the European background and Zürich's urban $CO_2$ dome was enhanced by an additional 15.4 ppm above its regional background (Figure 10). There was, however, a 12.1 ppm inter-site gradient among

the rooftop monitoring sites, (Figure 7; Table 3) and this resulted in $CO_2$ enhancements ranging from 7.9 to 20 ppm depending on what site was considered in isolation. Urban $CO_2$ enhancements of such magnitudes are comparable to other urban areas where similar analyses have been conducted and reported, for example, Briber et al. (2013); Xueref-Remy et al. (2023).

In Zürich's roadside environments, the proximity to the principal $CO_2$ emission source of traffic elevated $CO_2$ levels by an additional 26.4 ppm enhancement above Zürich's urban dome (Figure 10). This roadside enhancement, relative to other urban

environments, remained relatively constant during winter, spring, and summer, but it did increase in autumn. This suggests that the extra roadside loading of $CO_2$ remained mostly unchanged throughout the year and the regional background $CO_2$ was generally a more important driver of observed $CO_2$ in Zürich and, in turn, indicated that larger European scale source-sink processes drive much of the variability of $CO_2$ observed in Zürich's urban area. On average, the biogenic forced environments experienced higher $CO_2$ increments (28.5 ppm) than all other environments, including those located next to the roadside (Fig-




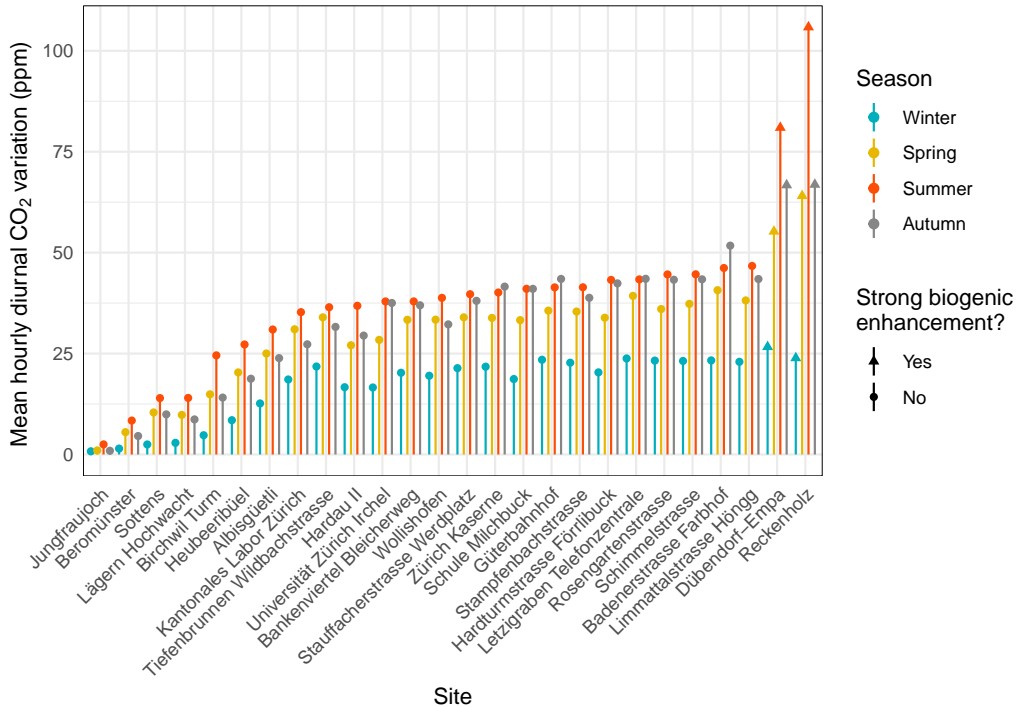

**Figure 9.** Mean diurnal ranges of $CO_2$ of minimum and maximum hourly means for four seasons for ZiCOS-M monitoring sites between July 2022 and July 2024.

ure 10). However, the large increments were only present in Zürch's growing seasons, especially in summer and autumn (34 and 36.4 ppm, respectively), and only in daily means, but not in daytime means as previously discussed.

### 3.2.4  Zürich's regional $CO_2$ background

A striking feature of the network behaviour was that Zürich's regional background $CO_2$ was highly variable. Figure 11 shows daily means of $CO_2$ for the Birchwil Turm and Lägern Hochwacht regional background monitoring sites and the high-alpine

site of Jungfraujoch. The figure shows the large dynamic range and pronounced temporal variability of the regional background enhancement relative to Jungfraujoch. Enhanced $CO_2$ was generally experienced in episodes with durations between four and 25 days, the longest of which was experienced in November and December 2022 (episode 6 in Figure 11). Here, an episode was defined as $CO_2$ being at least 14 ppm higher than Jungfraujoch for at least four sequential days. In total, 20 such episodes were identified during the monitoring period. $CO_2$ depletion events also occurred where Zürich's daily mean background $CO_2$

dropped below Jungfraujoch's, but this was rather rare and did not persist for longer than two days.

The 20 high $CO_2$ episodes, that were objectively classified, were explored and these episodes were not clearly related to wind speeds or wind conditions that would result in Zürich's or Winterthur's urban plumes nor emissions from Zürich's international airport being transported to the monitoring site. This suggested that Zürich's episodic high $CO_2$ background was driven by




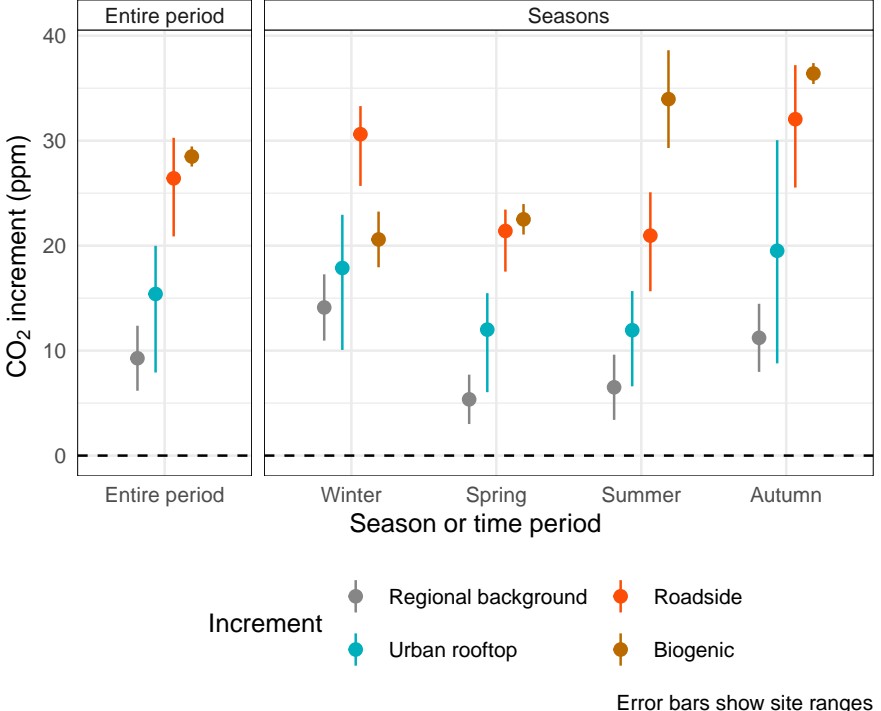

**Figure 10.** Daily $CO_2$ increments for different environments in Zürich between July 2022 and July 2024. The points indicate the groups' mean while the error bars show the minimum and maximum of the sites within the groups.

larger synoptic-scale processes when sink- and source-laden air masses pass over the region, as has been observed elsewhere

(Hurwitz et al., 2004; Pal et al., 2020; Davis et al., 2021). Furthermore, a key observation from this analysis is that Zürich's $CO_2$ is not driven by a simple anthropogenic loading on top of the hemispheric (or European) background, but rather it is a function of processes acting on different scales, all of which interact to produce ambient concentrations.

To investigate these 20 episodes further, FLEXible PARTicle dispersion model (FLEXPART) (Pisso et al., 2019) footprints for a European domain were run using Birchwil Turm as the receptor site for the time when the sensor network was operational.

This was done to determine where air masses were sourced from during the high $CO_2$ episodes. To clearly visualise the origin of the air masses experienced for each episode, each episode's surface was normalised by the entire period's mean in the same way as described in Sturm et al. (2013).

The FLEXPART footprints indicated that episodes occurred in most circulation regimes, with the exception of strong flows sourced from the Atlantic Ocean (six example episodes shown in Figure 12). Examples of northerly, southerly, western, and

455 eastern flows are present, as are conditions that are consistent with extended calm conditions. These footprints and the lack of clear correlation in wind behaviour demonstrate that high background $CO_2$ was influenced by larger scale processes, but high $CO_2$ episodes can occur in almost any circulation regime in Zürich.





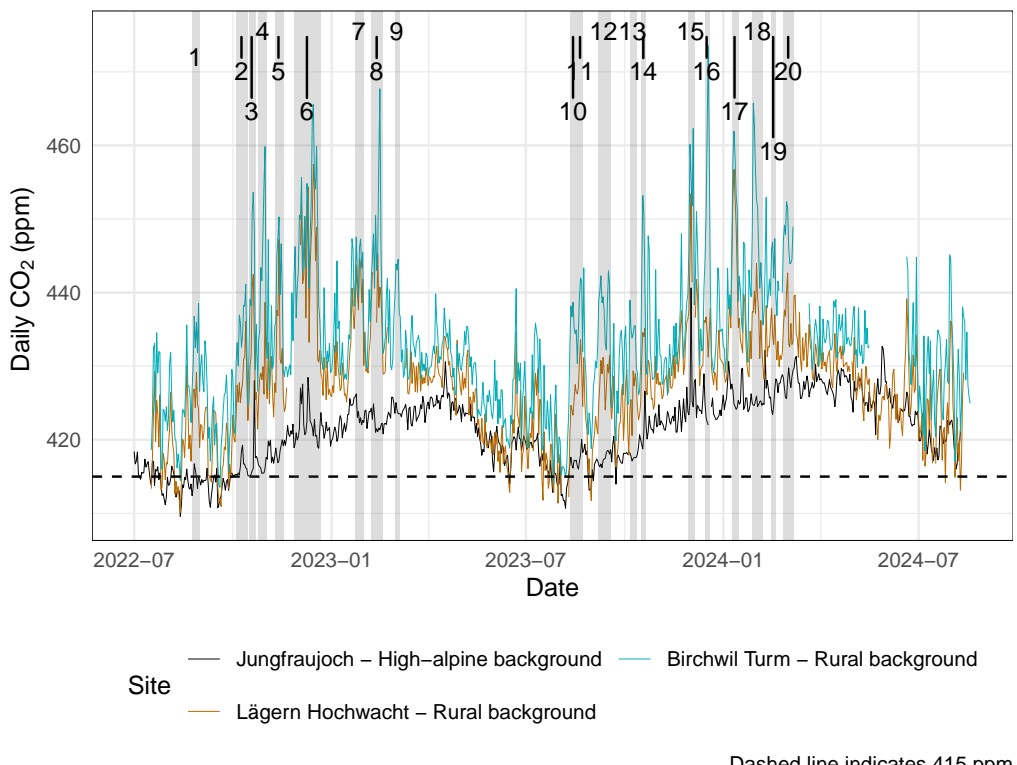

Dashed line indicates 415 ppm
Labelled shaded zones show the identified episodes

**Figure 11.** Daily $CO_2$ for two of ZiCOS-M's regional background sites and a high-alpine monitoring site between July 2022 and July 2024. High regional $CO_2$ episodes are indicated and labelled.

### 3.2.5 A low urban $CO_2$ event case study

The measurement performance of the sensors in the ZiCOS-M network was high enough to resolve local $CO_2$ source-sink

processes. An example of this was when $CO_2$ dropped below regional background concentrations across the city on March 3 2024. In Zürich on March 3, the highest temperatures of the year to that date were experienced, peaking at $18\,°C$. It represented one of the first days, if not the first day of 2024, when the biogenic uptake of $CO_2$ was strong due to photosynthesis. In the early afternoon, most sensor sites in the urban area experienced $CO_2$ levels well below those that were reported at the network's regional background monitoring sites, especially for five hours between 13:00 and 19:00 (Figure 13). The site that demonstrated

this plunge in $CO_2$ levels the clearest with a difference of 17 ppm was Universität Zürich Irchel, a rooftop monitoring site located in the northeast corner of the city (Table 3) and on the western border of the Zürichberg forest and hill (Figure 3).

An analysis of the wind behaviour at the time of low $CO_2$ concentrations revealed that the episode coincided with a wind direction shift to an east-southeast direction with wind speeds above $2\,\mathrm{m\,s^{-1}}$ (Figure 13). This strongly suggested that air from the Zürichberg forest that was depleted of $CO_2$ on the first growing day of the year was being transported over the Universität

Zürich Irchel site, and the city in general. The depleted $CO_2$ air over the city took another 24 hours to revert to the normal



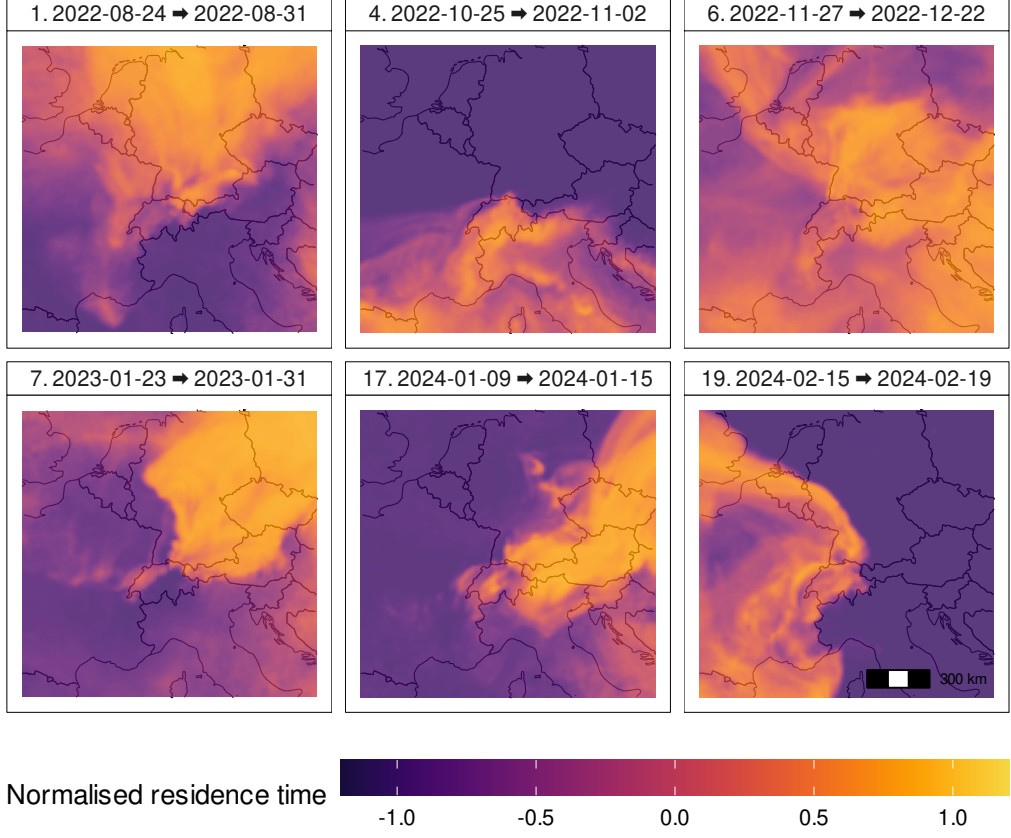

**Figure 12.** Normalised FLEXPART footprints (as described in Sturm et al. (2013)) for Birchwil Turm for six selected high $CO_2$ episodes experienced at the sensor network's regional background sites during the network's monitoring period.

situation where $CO_2$ in the urban area was higher than the regional background location. The Birchwil Turm monitoring site experienced $CO_2$ reduction too during the afternoon of March 3 because of biogenic uptake, but the specific and local process identified around Zürich's urban area did not affect this more distant location in the same way. This case study shows that the sensor network could provide insight into very specific and local-scale processes.

**4 Conclusions**

The ZiCOS-M $CO_2$ network's sensor performance and the insights gained into atmospheric processes influencing $CO_2$ while the network was operational have been presented and discussed. The measurement performance of the sensors was assessed through parallel measurements with a high-precision instrument under representative field conditions. After addressing ambient pressure, water vapour, and reference gas tests, the mean uncertainty was quantified at 0.98 ppm of RMSE for hourly mean

values. This level of measurement performance was high enough to confidently disentangle $CO_2$ gradients across Zürich's regional and urban areas.



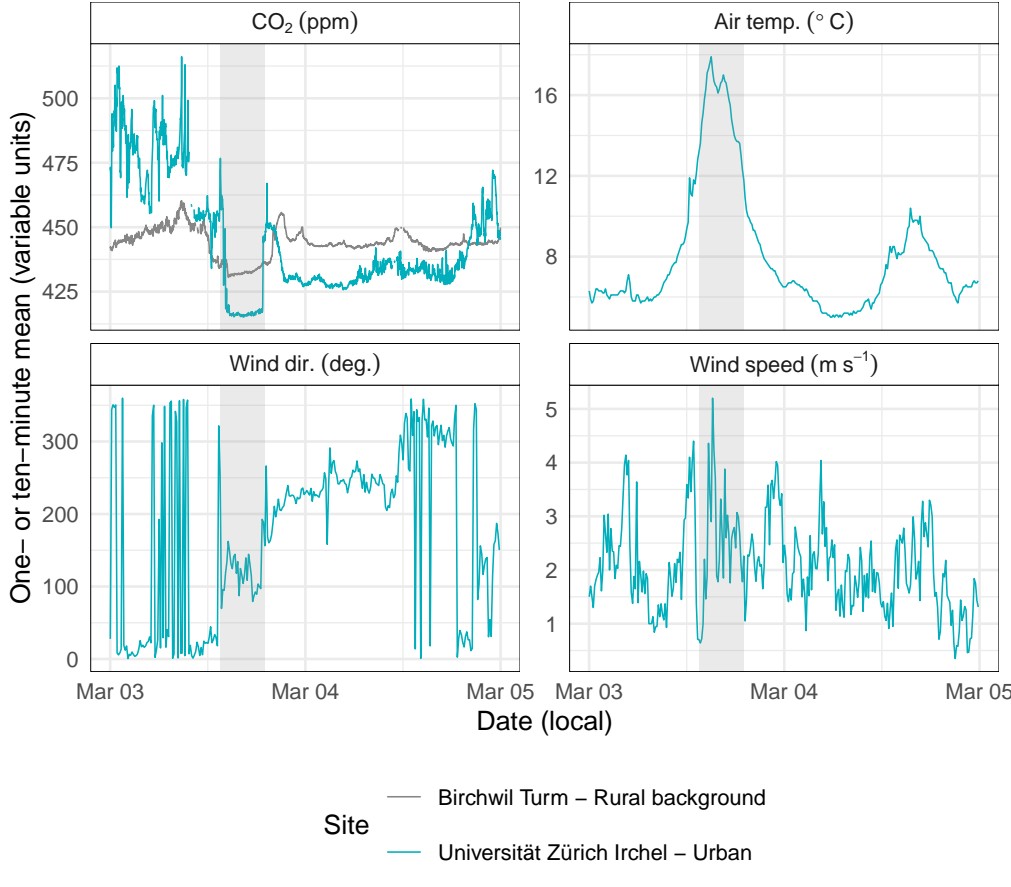

**Figure 13.** Time series of $CO_2$ and meteorological variables on March 3 and 4, 2024 demonstrating an episode where $CO_2$ at Universität Zürich Irchel dropped well below regional background levels due to forest air depleted of $CO_2$ passing over the monitoring site.

During the monitoring period between July 2022 and July 2024, the sites' $CO_2$ means ranged from 433 to 460 ppm, reflecting their different surrounding environments. The sites that experienced some of the highest $CO_2$ were strongly influenced by biogenic respiration with peak hourly $CO_2$ exceeding 730 ppm in the early hours of the morning. These levels were higher than 485 those found in Zürich's roadside environments but only reflected biogenic respiration in certain conditions and did not reflect the full $CO_2$ source-sink dynamics at these biogenic sites. Zürich's urban $CO_2$ dome was quantified, on average, as 15.4 ppm above the regional background and ranging between 7.9 and 20 ppm when considering the individual monitoring sites. Furthermore, the sensor network showed that the processes which drove $CO_2$ levels acted on multiple scales with synoptic-scale transport of $CO_2$-depleted or $CO_2$-enhanced air masses being especially important, and resulted in a very dynamic regional background that 490 experienced several high $CO_2$ episodes during the monitoring period. This illustrates that the observed $CO_2$ across the Zürich region was not a simple anthropogenic loading on top of a stable regional background, and emphasises the importance of



measurement sites placed in the surroundings of the city to characterise this background. These observations support previous studies, such as Turnbull et al. (2015).

The ZiCOS-M network provided important insights in its own right, but the observations will be used further in downstream
activities utilising atmospheric inversion modelling systems, including the ICON-ART (Schröter et al., 2018; Steiner et al., 2024) and GRAMM/GRAL models (Berchet et al., 2017), to determine the city's $CO_2$ emissions and compare them to the established approach based on the emission inventory (Stadt Zürich, 2023b, 2024). The sensor network also contained low-cost sensors that have not been presented here. However, details and the results gleaned from the low-cost sensors are in preparation. The ZiCOS-M network acts as a good example of an environmental gas sensing monitoring network in which the
measurement performance was adequate to answer a number of scientific research questions at an order of magnitude lower cost than would be possible with contemporary state-of-the-art $CO_2$ reference instruments.



**Appendices**

**Figures**

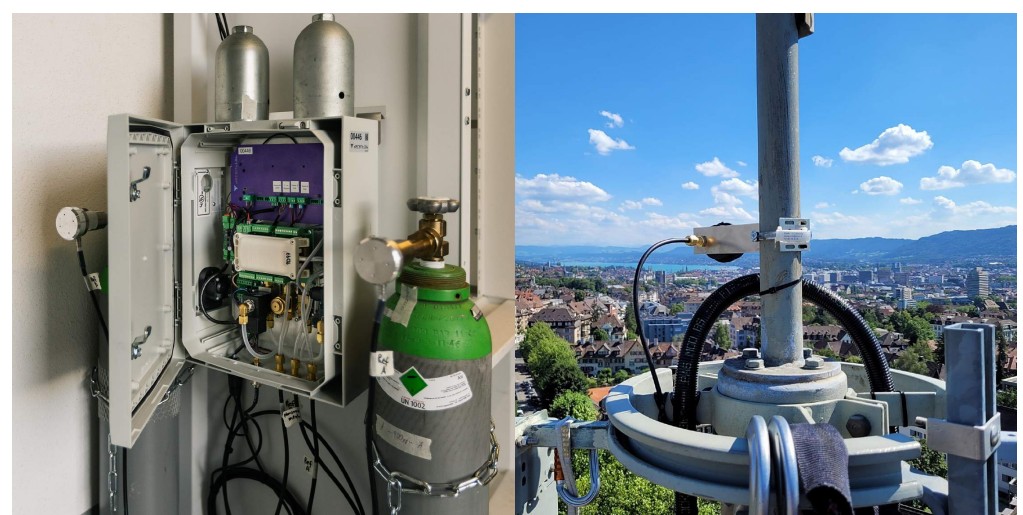

**Figure A1.** Examples of the $CO_2$ sensor's typical installations in the CARIN-ZH sensor network in Zürich city. The photograph on the left was taken by Pekka Pelkonen from ICOS RI.

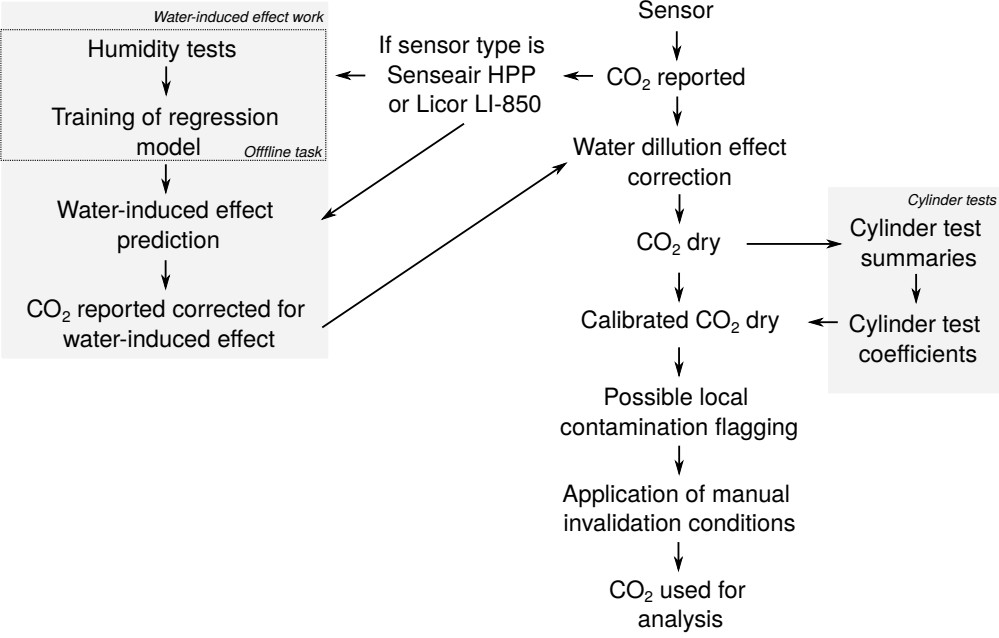

**Figure A2.** Schematic of the data handling steps for the sensors' $CO_2$ observations in the ZiCOS-M sensor network.





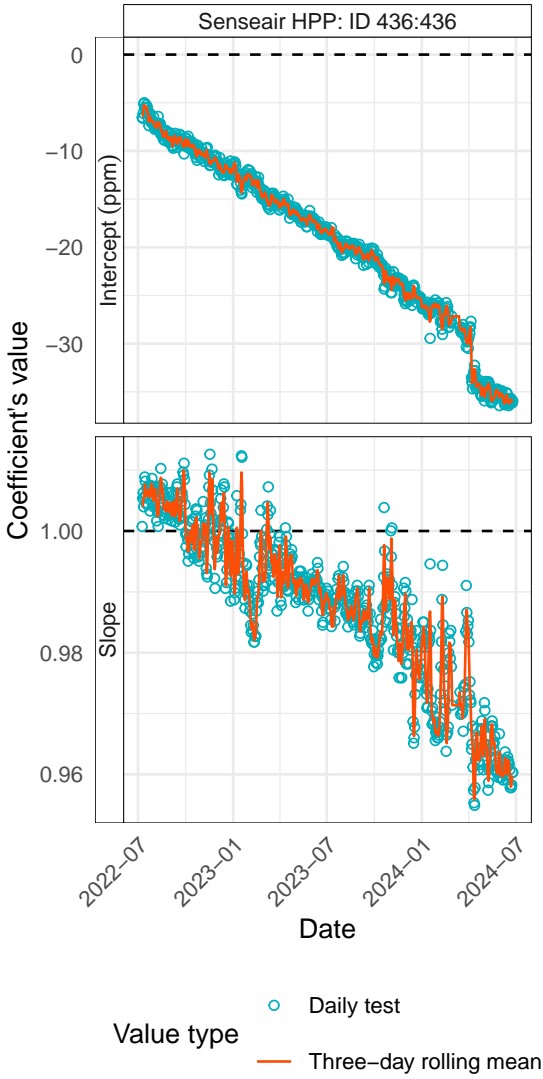

**Figure A3.** Slope and offset coefficients calculated from daily reference gas cylinder tests for an example sensor between July 2022 and July 2024. When applying the adjustment calculations, the three-day rolling mean coefficients were used. This particular sensor demonstrated a decline in baseline and sensitivity during the monitoring period.





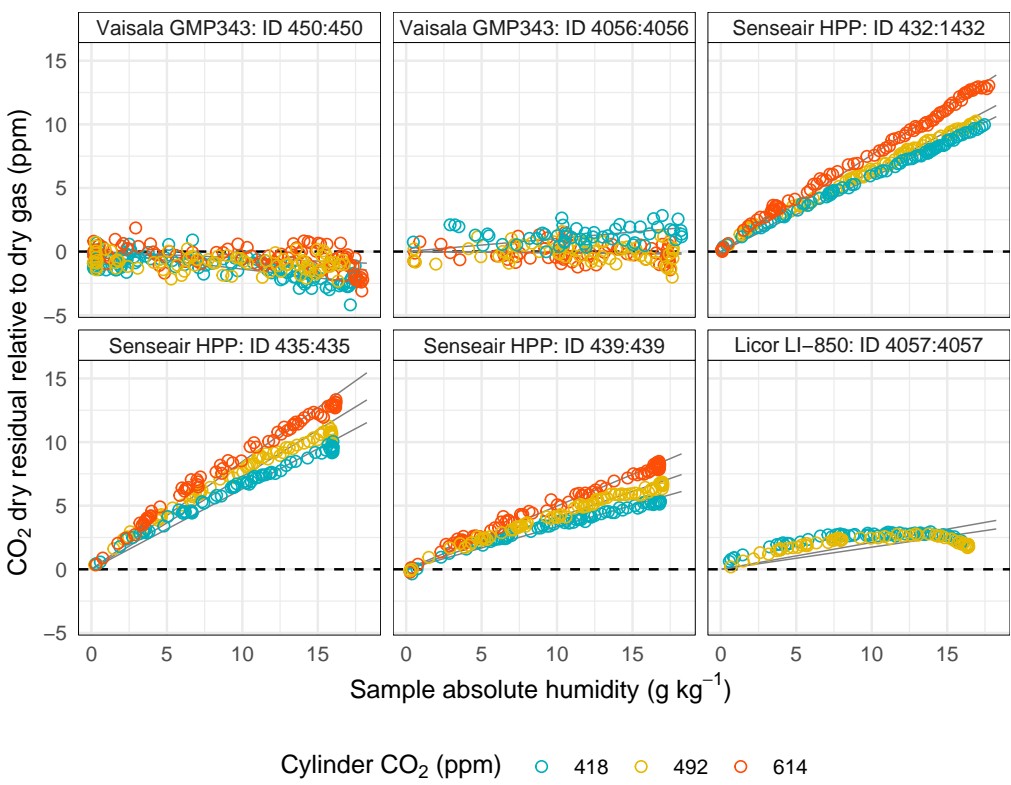

**Figure A4.** Examples of the water-induced response for three different types of $CO_2$ sensors during the decaying humidity tests. The Senseair HPP sensors strongly demonstrated the effect.



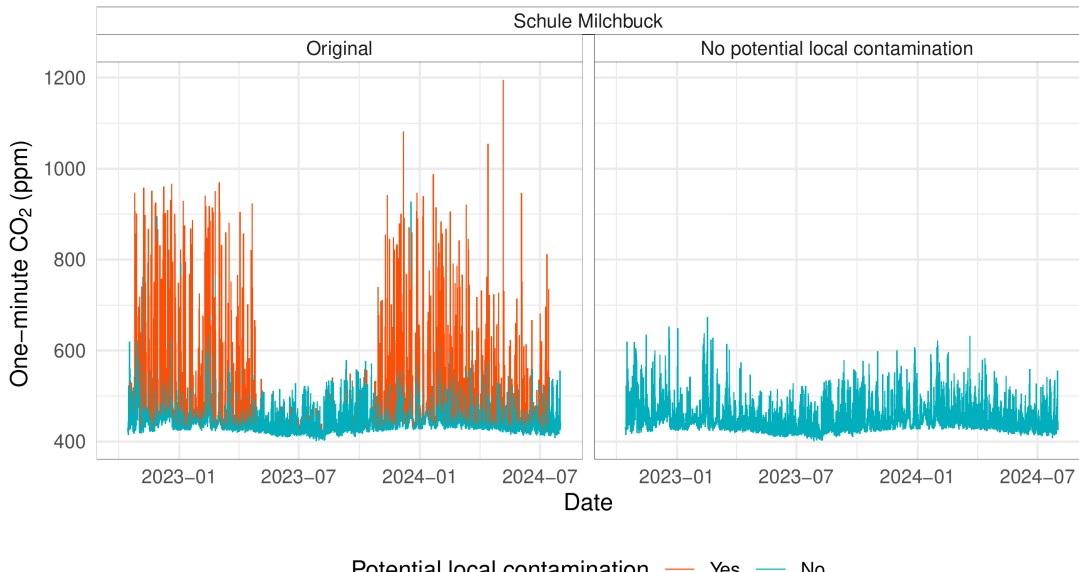

**Figure A5.** An example of the potential local contamination identification for a $CO_2$ sensor rooftop monitoring site between October 2022 and July 2024 where heating periods can be visually identified (this site is a school). The observations that were identified as contaminated by local contamination were flagged for optional handling by downstream data users.



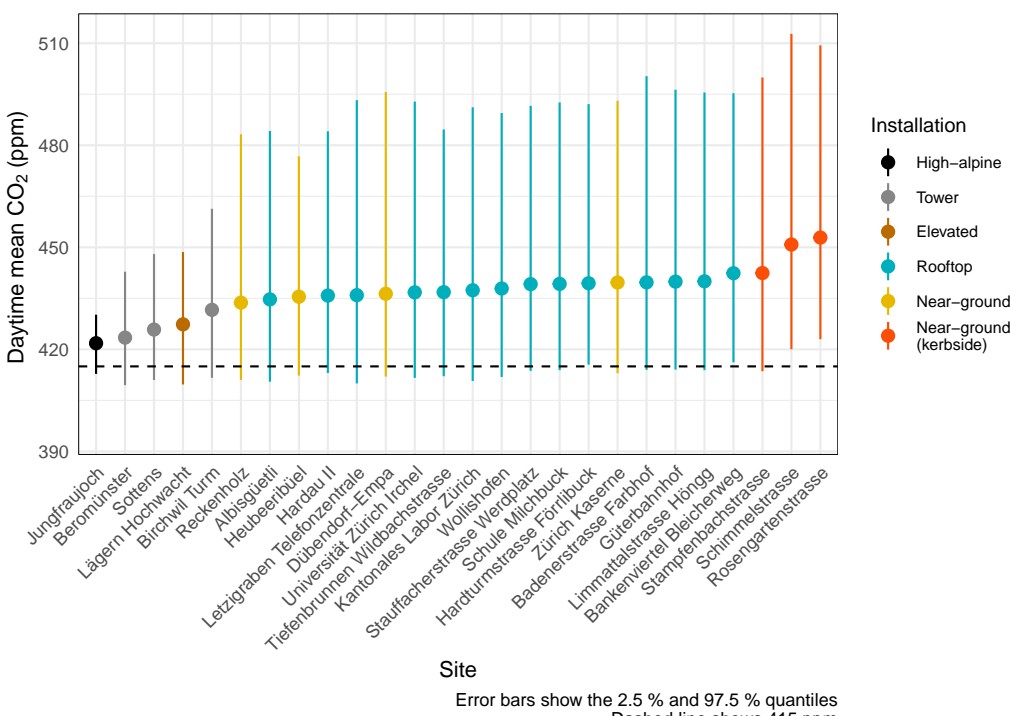

**Figure A6.** Mean daytime (local time hours between 10:00 and 16:00) $CO_2$ for ZiCOS-M's 26 monitoring sites between July 2022 and July 2024.



## Tables

**Table A1.** Fraction of observations that were identified as potentially locally contaminated with an outlier detector for 14 rooftop monitoring sites.

| Site | Installation | Fraction of potentially locally contaminated observations (%) |
|---|---|---|
| Hardturmstrasse Förrlibuck | Rooftop | 0.5 |
| Stauffacherstrasse Werdplatz | Rooftop | 0.5 |
| Hardau II | Rooftop (high) | 0.7 |
| Kantonales Labor Zürich | Rooftop | 0.7 |
| Letzigraben Telefonzentrale | Rooftop | 0.8 |
| Universität Zürich Irchel | Rooftop | 0.8 |
| Güterbahnhof | Rooftop | 0.9 |
| Bankenviertel Bleicherweg | Rooftop | 1.0 |
| Albisgüetli | Rooftop | 1.0 |
| Badenerstrasse Farbhof | Rooftop | 1.3 |
| Schule Milchbuck | Rooftop | 1.3 |
| Tiefenbrunnen Wildbachstrasse | Rooftop | 2.2 |
| Wollishofen | Rooftop | 2.6 |
| Limmattalstrasse Höngg | Rooftop | 5.1 |





**Table A2.** Hourly $CO_2$ dry air mole fraction error statistics of the $CO_2$ sensors when undergoing parallel measurements in field conditions with a high-precision gas analyser acting as a ground truth. The three error statistics units are ppm. The air temperature and absolute humidity are the means for the testing duration. Sensors 438 and 445 suffered from poor data transmission during testing so $n$ is significantly lower than the target of 336 hours (two weeks). The last row shows the means of the statistics.

| Sensor type | Sensor ID | Sensing element ID | Air temp. (°C) | Abs. hum. (g kg$^{-1}$) | $n$ | Mean bias | RMSE | MAE | $R^2$ |
|---|---|---|---|---|---|---|---|---|---|
| Senseair HPP | 429 | 3429 | 20.6 | 8.5 | 368 | 0.66 | 0.87 | 0.69 | 1.000 |
| Senseair HPP | 430 | 1430 | 16.8 | 10.9 | 406 | -0.23 | 1.49 | 1.13 | 0.999 |
| Senseair HPP | 433 | 433 | 19.6 | 11.7 | 649 | -0.18 | 0.82 | 0.66 | 1.000 |
| Senseair HPP | 434 | 1434 | 10.0 | 6.3 | 605 | -0.03 | 0.77 | 0.57 | 1.000 |
| Senseair HPP | 436 | 436 | 19.6 | 11.7 | 636 | -0.28 | 0.94 | 0.73 | 1.000 |
| Senseair HPP | 437 | 1429 | 13.3 | 8.9 | 339 | -0.04 | 0.89 | 0.69 | 1.000 |
| Senseair HPP | 438 | 437 | 5.0 | 6.2 | 127 | 0.01 | 0.46 | 0.34 | 1.000 |
| Senseair HPP | 439 | 439 | 6.0 | 6.3 | 320 | 0.19 | 1.48 | 1.08 | 0.997 |
| Senseair HPP | 441 | 429 | 6.0 | 6.3 | 315 | 0.07 | 1.36 | 1.03 | 0.998 |
| Senseair HPP | 443 | 443 | 9.2 | 7.5 | 531 | 0.22 | 1.33 | 1.01 | 0.998 |
| Senseair HPP | 444 | 2444 | 9.9 | 6.6 | 655 | -0.14 | 1.01 | 0.71 | 0.999 |
| Senseair HPP | 445 | 445 | 21.2 | 11.1 | 211 | -0.45 | 0.83 | 0.69 | 1.000 |
| Senseair HPP | 446 | 2429 | 9.0 | 7.5 | 489 | -0.20 | 1.03 | 0.80 | 0.999 |
| Vaisala GMP343 | 450 | 450 | 21.2 | 10.1 | 531 | -0.72 | 1.14 | 0.91 | 0.999 |
| Vaisala GMP343 | 451 | 451 | 5.2 | 6.0 | 1394 | -0.41 | 0.69 | 0.57 | 1.000 |
| Vaisala GMP343 | 452 | 452 | 5.9 | 5.7 | 311 | -0.67 | 0.94 | 0.80 | 1.000 |
| Vaisala GMP343 | 4060 | 4060 | 7.0 | 6.4 | 633 | -0.02 | 0.49 | 0.39 | 1.000 |
| Licor LI-850 | 4057 | 4057 | 16.8 | 10.9 | 415 | 0.60 | 1.07 | 0.95 | 1.000 |
| **Mean** | | | 12.3 | 8.3 | 496 | -0.09 | 0.98 | 0.76 | 0.999 |



**Equations**

The spectroscopic correction to address the sensors' water-induced response was conditionally applied to some sensor types in the following way:

$$CO_{2_\mathrm{w}} = \begin{cases} CO_{2_{reported}} - \left(CO_{2_{reported}} \cdot \beta_{CO_2} + w \cdot \beta_w\right) & \text{if sensor type was Senseair HPP or Licor LI-850} \\ CO_{2_{reported}} & \text{else} \end{cases} \tag{1}$$

Where $\beta_{CO_2}$ and $\beta_w$ represent slopes of multiple linear regression models that were trained on observations during humidity tests (Section 2.3.4). The Senseair HPP sensors' reported $CO_2$ required transformation to standard pressure and was done using standard atmospheric pressure (atm):

$$CO_{2_p} = \begin{cases} CO_{2_w} \cdot \left(\frac{1013.25}{P_o}\right) & \text{if sensor type was Senseair HPP} \\ CO_{2_w} & \text{if sensor type was Licor LI-850} \\ CO_{2_{reported}} & \text{else} \end{cases} \tag{2}$$

To compute $CO_2$ dry air mole fractions, water partial pressure ($P$) was calculated with the August-Roche-Magnus equation (Alduchov and Eskridge, 1997; Lawrence, 2005) using temperature ($T$) and relative humidity ($RH$) observations from the sensors' sample stream and, in turn, the water vapour mixing ratio was calculated, and the sensors' $CO_2$ moist air mole fractions were transformed using observed air pressure ($P_o$):

$$P = \frac{RH}{100} \cdot 6.1078 \cdot \exp\left(\frac{17.08085 \cdot T}{234.175 + T}\right) \tag{3}$$

$$CO_{2_{dry}} = \frac{CO_{2_p}}{(1 - \frac{P}{P_o})}, \tag{4}$$

Finally, $CO_2$ dry air mole fractions were calibrated with slopes ($\beta_{cylinder}$) and offsets ($\alpha_{cylinder}$) calculated from reference gas tests conducted every 25 hours:

$$CO_{2_{dry\,cal}} = CO_{2_{dry}} \cdot \beta_{cylinder} + \alpha_{cylinder}. \tag{5}$$

The product of these transformations was $CO_2$ dry air mole fractions calibrated to the WMO X2019 calibration scale (Hall et al., 2021).



*Data availability.* The data sources used in this work are described and the observations are available via the ICOS Cities data portal
(`https://citydata.icos-cp.eu/portal`). The hourly field and intercomparison observations used for the analysis are also publicly accessible in a persistent data repository (Grange, 2024a, `https://doi.org/10.5281/zenodo.13759332`). Additional data
and information are available from the authors upon reasonable request.

*Author contributions.* SKG conceived the research questions, conducted the data analysis, and wrote the manuscript with assistance from
all authors. PR, AF, DB, CH, and LE designed the sensor network, were project managers, and contributed to data analysis activities. PR led
the sensor network's installation and on-site maintenance. All authors contributed to revising and improving the manuscript.

*Competing interests.* The authors declare no competing interest.

*Acknowledgements.* This work was funded by the European Union's Horizon 2020 research and innovation programme, grant agreement
number 101037319, named Pilot Applications in Urban Landscapes - towards integrated city observatories for greenhouse gases (PAUL) and
is known as ICOS Cities. The project team thanks the collaborators from Umwelt- und Gesundheitsschutz Zürich (UGZ; the environmental
department of Zürich city) and Swisscom for their help regarding sensor sites and installations. Decentlab GmbH is thanked for their development of sensor hardware and for developing reliable data transmission and storage infrastructure. Simone Baffelli and Lucas Fernandez
Vilanova are thanked for their contributions to legacy software systems as is Nikolai Ponomarev for providing feedback on the sensors'
observations. Beat Schwarzenbach and Stephan Henne are thanked for allowing easy access to three additional sites' $CO_2$ observations from
the NABEL and ICOS-CH monitoring networks. Stephan Henne is thanked for a second time for his running of the FLEXPART footprints.
The sensor network maintenance was only possible with the help of civil servants (*Zivis*) when the network was in operation. The Zivis
were Wisnu Lang, Simon Rohrbach, Davide Bernasconi, Hannes Wäckerlig, Michael Kovac, Josua Stoffel, Urban Brunner, Ulysse Schaller,
Leonardo Beltrami (not a civil servant but a visiting student), Stefan Lampart, Yann von der Weid, Jan Krummenacher, and Quirin Beck;
many thanks for your contributions. Finally, the two other ICOS Cities sensor network groups in Munich and Paris are thanked for their
fruitful collaboration.



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
