# Peer review of "The ZiCOS-M CO2 sensor network: measurement performance and CO2 variability across Zürich"

_EGUsphere, 2024_

## Author Comment (AC1)

**Response to reviewers for** EGUSPHERE-2024-2925 (*The ZiCOS-M CO$_2$ sensor network: measurement performance and CO$_2$ variability across Zürich*)

December 14, 2024

Stuart K. Grange[*], Pascal Rubli, Andrea Fischer, Dominik Brunner, Christoph Hueglin, and Lukas Emmenegger

[*]s.k.grange@gmail.com

**Anonymous Referee #1**

**General comments**

This is an extremely careful, thorough description of the performance of an urban sensor network. It was a pleasure to read. I recommend publication.

Many thanks for your very positive comments. We are happy that you enjoyed reading our manuscript, please find our responses to your itemized comments below.

**Itemised comments**

I have two minor points that could be clarified.

1. Line 228–230: It would be useful to confirm that a longer smoothing and diurnal variation in the calibration time has no effect on the analysis.

   The smoothing approach for the application of the calibration coefficients generated by the reference gas tests was tested thoroughly during the parallel measurements programme. We believe the full analysis of gas test frequency and how to handle the tests' coefficients is outside the scope of the manuscript because it was an internal work package to optimise our sensor operations. However, an example of a sensitivity analysis is below (Figure 1). Figure 1 shows the lowest amount of error was generally found with daily gas tests and a three-day rolling mean being applied to the gas tests' coefficients, thus, these criteria were chosen for the sensor operations.

2. Table 3: Does the lower data capture at a handful of sites (Beromunster, for example) mean the data from this site is not sampled across seasons with the same distribution as other sites?

   Beromüster was the only site that suffered from a significant gap in its time series (between late August 2023 and early February 2024) and thus the monitoring record is slightly biased for this site. A sentence has been added stating this issue to the readers:

   "Beromünster suffered an observation gap between late August 2023 and early February 2024 due to instrument failure and therefore, had a lower data capture rate compared to the other sites."

[Figure]

Figure 1: Sensitivity of $CO_2$ measurement error when the frequency of reference gas cylinder tests and number of days the calibration coefficients were smoothed during parallel measurements for a single sensor.

**Anonymous Referee #2**

**General comments**

This manuscript describes instrument performance and results from a network of nondispersive infrared $CO_2$ sensors that are lower cost than the typical instruments used for studying urban greenhouse gas emissions. The advantages of a viable lower cost instrument are many — including enhancing the ability to spatially resolve emissions within a city. With increased spatial resolution we can also learn about the sectoral contributions to emissions and thus better advise policymakers as to the most effective means of mitigation. New instrumentation must be tested for compatibility and this manuscript does so for 26 sites (including three instrument types, but mostly Senseair HPP sensors). The research presented is well thought-out and the manuscript well-written. I recommend publication after minor revisions.

Many thanks for your positive comments. Please find our responses to your itemized comments below.

**Itemised comments**

**Minor**

1. Abstract: I don't think the mean bias being -0.09 ppm is relevant. The only thing we actually care about is that the range of biases for the instruments varied from -0.72 to +0.66 ppm. I suggest rewording. It would also be good to specify that these results came from co-locations of the instruments with a reference instrument of periods of two weeks or more. For a while I was thinking that the statistics came from co-locations of four mid-cost sensors at the four sites with reference instruments, in which case the statistics would not be nearly as representative.

   We do not totally agree that the average mean bias is irrelevant. We would argue that the result of having symmetrical biases for the sample of sensors is a good outcome. However, the abstract has been edited to only state the range of the mean bias because the average mean bias information is discussed in the manuscript. The abstract now also states that the performance statistics were calculated for a period of two weeks or more.
   "...while the mean bias ranged between -0.72 and 0.66 ppm when undergoing parallel measurements with a high-precision reference gas analyser for a period of two weeks or more."

2. Abstract: Consider adding the daytime mean gradient of 6.5 ppm (half of 13 ppm) to provide context for the attained compatibility.

   We agree that the inclusion of the network's daytime gradient would be beneficial but the abstract's word count is very tight and we do not believe this result is critical for the manuscript's

main argument. Daytime gradients are more useful for modelling activities and these results are discussed in the results section.

3. Pg. 8 Section 2.3: This phrase is worrisome: "adjustment strategies to increase their agreement with observations generated by reference instrumentation" This procedure sounds odd, because it is the metric by which the instruments are evaluated. We need some details here.

We have removed this statement that was within parentheses. As mentioned above, improvements to the accuracy of observations, by definition, will increase their agreement to those generated by reference instruments and therefore, was a redundant statement.

4. It's not until the results that we learn that there were parallel measurements at the Dubendorf site for at least two weeks for each of the instruments. This should be in the methods as well, and described more clearly in the abstract. (There is a single phrase indicating parallel measurements in section 2.3.1 but the idea that all instruments were tested for two weeks or more is important.)

The abstract has been edited as part of the revision for point 1 above. The methods have been edited to explain the two-week-long intercomparison exercises at Dübendorf-Empa.

"The network's testing site, Dübendorf-Empa, is also an air quality morning site that is part of the National Air Pollution Monitoring Network (NABEL). Dübendorf-Empa was used for intercomparison exercises and model training by using the reference $CO_2$ time series provided at this site. The intercomparisons conducted at Dübendorf-Empa allowed for the $CO_2$ sensors to be run in parallel with a high-precision gas analyser in ambient conditions for at least two weeks for a representative and robust testing procedure. Therefore, the measurement performance presented in Section 3.1 is highly representative of what was achieved during operational monitoring."

5. Figure 4: The text indicates that the discontinued Senseair HPP sensor often performed more poorly than the other sensors. I think the most important statistic is the mean bias and the mean bias is better for the Senseair, overall (maybe too few of the others to really say).

When discussing Figure 4, we focused on the root mean square error (RMSE) metric because it includes both variance and bias dimensions, thus, the Senseair HPP sensor performed slightly worse than the other sensor models that were tested. We purposefully do not discuss these patterns too much because the sample size of sensors would need to be much larger to be confident that the patterns observed are indeed true. The final sentence of this section has been edited to state this explicitly:

"However, because of the irregular number of each sensor type, additional sensor units would need to be used to confirm these apparent patterns and not overinterpret the rather small

differences observed among the different sensor models."

**Technical**

1. Pg. 2, first full paragraph. I don't know if I'd say that the monitoring of $CO_2$ in urban areas has not been a priority. I understand the idea (there are more measurements of air pollutants), but there has been quite a lot of work on it!

   We agree. The thrust of this sentence was to relate the gap between urban air quality monitoring and urban $CO_2$ monitoring, but as mentioned, plenty of work has been done to address this. We have edited the sentence to clarify this:

   "The monitoring of $CO_2$ in urban areas has been a lesser priority when compared to traditional air pollutants because of the lack of legal standards for $CO_2$."

2. Pg. 2, line 25: complementary instead of complimentary

   Thanks for catching this error, the word has been replaced.

3. General: I don't know about British English but American English we'd say either in "Zurich" or in "the city", but not in "Zurich city".

   We feel the use of "Zürich city" is appropriate in this context. Admittedly, Zürich city can sound clunky in some situations, we use this term because Zürich is the English name of the canton (state), region, and the City of Zürich. We use the specific Zürich city term to avoid confusion among these three geographic regions.

4. Pg. 2, first full paragraph. Missing references for the US high precision network (e.g., LA, NorthEast Corridor, Indianapolis)

   These three networks have been added to the paragraph along with citations:
   "Prominent urban $CO_2$ monitoring networks include the Berkeley Environmental Air-quality and $CO_2$ Network (BEACO$_2$N) located across the San Francisco Bay area (Shusterman et al., 2016; Turner et al., 2016; Kim et al., 2018; Delaria et al., 2021), the Indianapolis (INFLUX) Urban Test Bed (Turnbull et al., 2015; Davis et al., 2017), the Los Angeles Megacity Carbon Project (Verhulst et al., 2017), the Northeast Corridor tower network (Karion et al., 2020)..."

5. Pg.4, last paragraph: Please include manufacturers and part numbers. I'm particularly interested in the demand-flow regulator, but it's good to have all the details documented.

   A new table in the Appendix (Table A1) has been added with the makes and models of the principal parts/components of the sensor packages. This table is referenced on Page 4 and the schematic of the sensor package (Figure 1).

6. Figure 1: The pump is upstream of the $CO_2$ sensor and thus the measurements will be affected by leaks within the pump or connections. Please discuss this choice. Why are there RH/T sensors both upstream and downstream of the $CO_2$ sensor?

The location of the pump, upstream of the $CO_2$ sensor, was selected to obtain the same (ambient) pressure for the sample and the reference gases. If the pump was placed downstream, then the dimensions of the sampling tube and the pressure drop at the sample filter would have an effect on the gas pressure. However, in our (upstream) design, the pump must be leak-free. Small leaks would be somewhat compensated by the fact that the reference gases are also added before the pump and are thus affected similarly as the sample. Furthermore, a leaking sampling system or pump was identified very rapidly when the samples of the reference gases were found to be not dry and such tests were a component of the network operations.

All sensors were equipped with the upstream temperature and relative humidity sensors that were installed in the sample line. These are what are referred to as the "ancillary sensors in the gas stream" in the manuscript and were used to measure the conditions of the samples. The downstream temperature and relative humidity sensors were only installed on some sensors and were designed to measure enclosure/sensor package temperature, relative humidity, and pressure. The pressure measurements for these sensors were important for the dilution corrections.

7. Table 2: A column indicating the instrument type would be a nice addition to Table 2.

A new column has been added to Table 2 indicating the monitor type for each site.

8. Table 3: How is near-ground difference from near-ground (kerbside)?

The near-ground and near-ground (kerbside) sensor installations were installed at comparable heights, but the kerbside installations were designed to be very close to rather busy streets. The installation of the kerbside sensors was more complicated because the sensor installation and inlet routing had to conform to the limited space at these sites which was unlike the other ground-level sites.

9. Pg. 18, line 365: What is the landcover surrounding Sottens (e.g., within 10 km)? Agriculture?

Yes, Sottens is situated in an agricultural area. There are a few pockets of forest, small villages, and a handful of minor roads. The surrounding landuses of Sottens and Birchwil Turm are very much comparable and almost certainly not be responsible for the differences observed between these two Swiss plateau sites.

**Other changes**

1. An additional funding agency and grant number has been added to the acknowledgments: "ICOS Switzerland (ICOS-CH), Swiss National Science Foundation, grant 20F 120_19822."